# Learning Pattern-Specific Experts for Time Series Forecasting Under Patch-level Distribution Shift

**Yanru Sun, Zongxia Xie,[*] Emadeldeen Eldele[‡†], Dongyue Chen, Qinghua Hu, Min Wu[‡]\***
College of Intelligence and Computing, Tianjin University, China
[‡]I2R, Agency for Science, Technology and Research, Singapore.
[†]Department of Computer Science, Khalifa University, UAE.
{yanrusun, dyuechen, huqinghua}@tju.edu.cn, caddiexie@hotmail.com,
emad0002@ntu.edu.sg, wumin@a-star.edu.sg

## Abstract

Time series forecasting, which aims to predict future values based on historical data, has garnered significant attention due to its broad range of applications. However, real-world time series often exhibit heterogeneous pattern evolution across segments, such as seasonal variations, regime changes, or contextual shifts, making accurate forecasting challenging. Existing approaches, which typically train a single model to capture all these diverse patterns, often struggle with the pattern drifts between patches and may lead to poor generalization. To address these challenges, we propose **TFPS**, a novel architecture that leverages pattern-specific experts for more accurate and adaptable time series forecasting. TFPS employs a dual-domain encoder to capture both time-domain and frequency-domain features, enabling a more comprehensive understanding of temporal dynamics. It then performs subspace clustering to dynamically identify distinct patterns across data segments. Finally, these patterns are modeled by specialized experts, allowing the model to learn multiple predictive functions. Extensive experiments on real-world datasets demonstrate that TFPS outperforms state-of-the-art methods, particularly on datasets exhibiting significant distribution shifts. The data and code are available: https://github.com/syrGitHub/TFPS.

## 1 Introduction

Time series forecasting plays a critical role in various domains, such as finance [18], weather [3, 63, 24], traffic [35, 22], and others [59, 33, 68], by modeling the relationship between historical data and future outcomes. However, the inherent complexity of time series data, including temporal dependencies and non-stationarity, poses significant challenges in achieving reliable forecasts.

Recent Transformer-based models have shown great promise in time series forecasting due to their ability to model long-range dependencies [29, 53]. In particular, models like PatchTST [44] split continuous time series into discrete patches and process them with Transformer blocks. While these models are effective, a closer examination reveals that patches often exhibit distribution shifts, which are frequently associated with concept drift [36]. For example, patches from different regimes, seasons, or operating modes may not only differ in statistical properties [27], but also in the functional relationships between historical and future values [60, 56]. However, this variability contradicts the assumptions of most existing models [44, 69, 9], which adopt the Uniform Distribution Modeling (UDM) strategy by treating all patches as samples from a single underlying distribution. This oversimplified view ignores structural heterogeneity and temporal variation across segments, thereby limiting the model's ability to generalize and degrading its forecasting performance [43, 25].

---

[*]Corresponding author

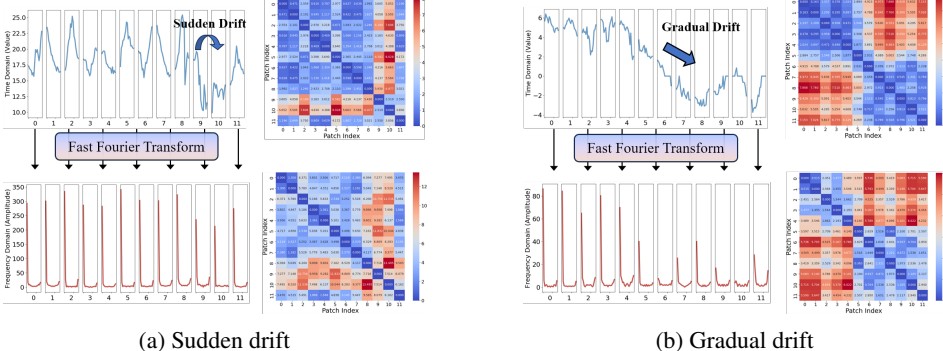

|(a) Sudden drift|(b) Gradual drift|

Figure 1: Illustration of distribution shifts between time series patches on the ETTh1 dataset, quantified by Wasserstein distance. The combined time- and frequency-domain views reveal richer and more complementary shift patterns arising from temporal non-stationarity.

To quantify these distributional shifts, we split the ETTh1 dataset into patches and analyze two representative cases: sudden drift and gradual drift, in both time and frequency domains. Specifically, we compute the Wasserstein distance between patches and visualize the results as heatmaps in Figure 1, which clearly illustrate the discrepancies across segments. Notably, sudden drift (Figure 1 (a)) leads to a sharp discrepancy between patches 9 and 10 and the remaining segments, while gradual drift (Figure 1 (b)) reveals that patches 0 to 5 differ from patches 6 to 11, exhibiting a progressive shift that makes forecasting more challenging. Furthermore, the frequency domain offers a complementary perspective on the shifts [42]. These observations highlight the complex and evolving nature of time series data, where different segments may follow distinct distributions and exhibit heterogeneous temporal patterns [55, 27, 17, 56, 52].

To address the challenges posed by distribution shifts in time series data, we propose a novel **T**ime-**F**requency **P**attern-**S**pecific (**TFPS**) architecture to effectively model the complex temporal patterns. In particular, TFPS consists of the following three key components. The first is a Dual-Domain Encoder (DDE), which extracts features from both time and frequency domains to provide a comprehensive representation of the time series data, enabling the model to capture both short-term and long-term dependencies. Second, TFPS addresses the issue of concept drift by incorporating a Pattern Identifier (PI), that utilizes a subspace clustering approach to dynamically identify the distinct patterns across patches. This enables the model to effectively handle nonlinear cluster boundaries and accurately assign patches to their corresponding clusters. Finally, TFPS constructs a Mixture of Pattern Experts (MoPE)—a set of specialized expert models, each tailored to a specific pattern identified by the PI. By dynamically assigning patches to the appropriate experts, TFPS learns pattern-specific predictive functions that effectively capture heterogeneous temporal dynamics and distributional variations. This specialized modeling strategy enhances the model's adaptability and yields significant forecasting improvements, particularly on datasets with severe distributional drift.

In summary, the key contributions of this work are:

- We introduce a novel pattern-specific forecasting paradigm that enables segment-wise expert modeling based on latent pattern structure, overcoming the limitations of uniform modeling under distribution shift.

- We propose TFPS, a dual-domain framework that integrates time- and frequency-domain representations with subspace clustering and dynamic expert routing, enabling the model to explicitly adapt to concept drift and capture evolving patterns in non-stationary time series.

- We evaluate our approach on nine real-world multivariate time series datasets, demonstrating its effectiveness. Our model achieves top-1 performance in 57 out of 72 settings, showcasing its competitive edge in improving forecasting accuracy.

## 2 Related Work

**Time Series Forecasting Models.** In recent years, deep models with elaborately designed architectures have achieved great progress in time series forecasting [51, 67, 28, 45]. Approaches like TimesNet [61] and ModernTCN [41] utilize convolutional neural networks with time-series-specific modifications, making them better suited for forecasting tasks. Additionally, simpler architectures such as Multi-Layer Perceptron (MLP)-based models [69, 8] have demonstrated competitive performance. However, Transformer-based models have gained particular prominence due to their ability to model long-term dependencies in time series [70, 62, 71, 29]. Notably, PatchTST [44] has become a widely adopted Transformer variant, introducing a channel-independent patching mechanism to enhance temporal representations. This approach has been further extended by subsequent models [29, 9].

While previous work has primarily focused on capturing nonlinear dependencies in time series through enhanced model structures, our approach addresses the distribution shifts caused by evolving patterns within the data, which is a key limitation of existing methods.

**Non-stationary Time Series Forecasting.** Non-stationarity in time series data complicate predictive modeling, necessitating effective solutions to handle shifting distributions [36, 11]. To address varying distributions, normalization techniques have emerged as a focal point in recent research, aiming to mitigate non-stationary elements and align data with a consistent distribution.

For instance, adaptive norm [46] applies z-score normalization using global statistics and DAIN [47] introduces a neural layer for adaptively normalizing each input instance. Reversible instance normalization (RevIN) [20] is proposed to alleviate series shift. Furthermore, Non-stationary transformer [31] points that directly stationarizing time series will damage the model's capability to capture specific temporal dependencies and introduces an innovative de-stationary attention mechanism within self-attention frameworks. Recent advancement include Dish-TS [10], which identifies both intra- and inter-space distribution shifts in time series data, and SAN [34], which applies normalization at the slice level, thus opening new avenues for handling non-stationary time series data. Lastly, SIN [17] introduces a novel method to selecting the statistics and learning normalization transformations to capture local invariance in time series data.

However, normalization methods can only address changes in statistical properties, and over-reliance on them may lead to over-stationarization, where meaningful temporal variations are inadvertently smoothed out [34]. In contrast, our approach preserves the intrinsic non-stationarity of the original series in the latent representation space, enabling the model to better adapt to evolving regimes by tailoring experts to diverse temporal patterns and distributional structures.

## 3 Method

### 3.1 Preliminaries

Time series forecasting aims to uncover relationships between historical time series data and future data. Let $\mathcal{X}$ denote the time series, and $x_t$ represent the value at timestep $t$. Given the historical time series data $X = [x_{t-L+1}, \cdots, x_t] \in \mathbb{R}^{L \times C}$, where $L$ is the length of the look-back window and $C > 1$ is the number of features in each timestep, the objective is to predict the future series $Y = [x_{t+1}, \cdots, x_{t+H}] \in \mathbb{R}^{H \times C}$, where $H$ is the forecast horizon.

### 3.2 Overall Architecture

Our model introduces three novel components: the Dual-Domain Encoder (DDE), the Pattern Identifier (PI), and the Mixture of Pattern Experts (MoPE), as illustrated in Figure 2. The DDE goes beyond traditional time-domain encoding by incorporating a frequency encoder that applies Fourier analysis, transforming time series data into the frequency domain. This enables the model to capture periodic patterns and frequency-specific features, providing a more comprehensive understanding of the data. The PI is a clustering-based module that distinguishes patches with distinct patterns, effectively addressing the variability in the data. MoPE then utilizes multiple MLP-based experts, each dedicated to modeling a specific pattern, thereby enhancing the model's ability to adapt to the temporal dynamics of time series. Collectively, these components form a cohesive framework that effectively handles concept drift between patches, leading to more accurate time series forecasting.

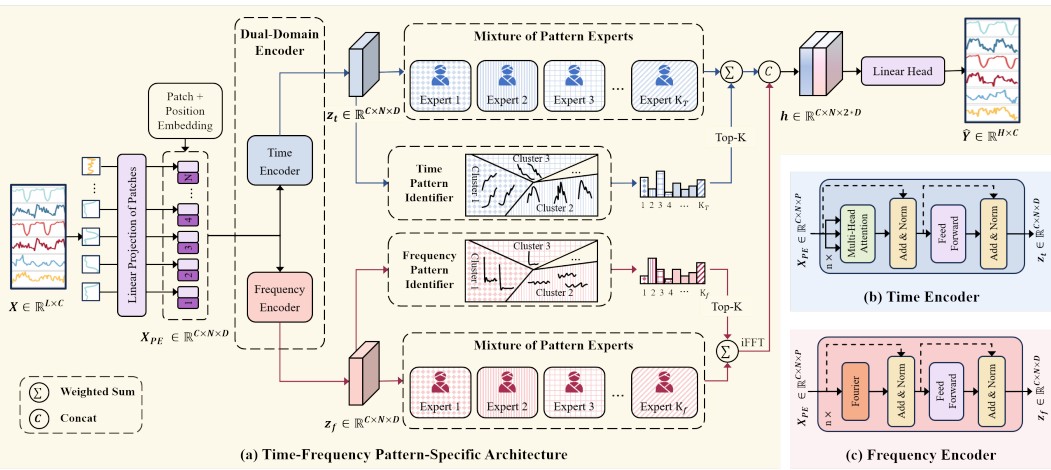

Figure 2: The structure of our proposed TFPS. The input time series is divided into patches, and positional embeddings are added. These embeddings are processed through two branches: time-domain branch and frequency-domain branch. Each branch consists of three key components: (1) an encoder to capture patch-wise features, (2) a clustering mechanism to identify patches with similar patterns, and (3) a mixture of pattern experts block to model the patterns of each cluster. Finally, the outputs from both branches are combined for the final prediction.

### 3.3 Embedding Layer

Firstly, the input sequence $X \in \mathbb{R}^{L \times C}$ is divided into patches of length $P$, resulting in $N = \lfloor \frac{(L-P)}{S} + 2 \rfloor$ tokens, where $S$ denotes the stride, defining the non-overlapping region between consecutive patches. Each patch is denoted as $\mathcal{P}_i \in \mathbb{R}^{C \times P}$. These patches are then projected into a new dimension $D$, via a linear transformation, such that, $\mathcal{P}_i \to \mathcal{P}'_i \in \mathbb{R}^{C \times D}$.

Next, positional embeddings are added to each patch to preserve the temporal ordering disrupted during the segmentation process. The position embedding for the $i$-th patch, denoted as $E_i$, is a vector of the same dimension as the projected patch. The enhanced patch is computed by summing the original patch and its positional embedding: $X_{PE_i} = \mathcal{P}'_i + E_i$, and $X_{PE} = \{X_{PE_1}, X_{PE_2}, \cdots, X_{PE_N}\}$. Notably, the positional embeddings are learnable parameters, which enables the model to capture the temporal dependencies in the time series more effectively. As a result, the final enriched patch representations are $X_{PE} \in \mathbb{R}^{C \times N \times D}$.

### 3.4 Dual-Domain Encoder

As shown in Figure 1, both time and frequency domains reveal distinct concept drifts that can significantly affect the performance of forecasting models. To effectively address these drifts, we propose a Dual-Domain Encoder (DDE) architecture that captures both temporal and frequency dependencies inherent in time series data.

We utilize the patch-based Transformer [44] as an encoder to extract embeddings for each patch, capturing the global trend feature. The multi-head attention is employed to obtain the attention output $\mathbf{O}_t \in \mathbf{R}^{N \times D}$:

$$\mathbf{O}_t = \text{Attention}(Q, K, V) = \text{Softmax}\left(\frac{QK^T}{\sqrt{d_k}}\right) V, \tag{1}$$
$$Q = X_{PE}\mathbf{W}_Q, \quad K = X_{PE}\mathbf{W}_K, \quad V = X_{PE}\mathbf{W}_V.$$

The encoder block also incorporates BatchNorm layers and a feed-forward network with residual connections, as shown in Figure 2 (b). This process generates the temporal features $z_t \in \mathbb{R}^{C \times N \times D}$.

In parallel with the time encoder, we incorporate a Frequency Encoder by replacing the self-attention sublayer of the Transformer with a Fourier sublayer [26]. This sublayer applies a 2D Fast Fourier Transform (the number of patches, hidden dimension) to the patch representation, expressed as:

$$\mathbf{O}_f = \mathcal{F}_{patch}(\mathcal{F}_h(X_{PE})). \tag{2}$$

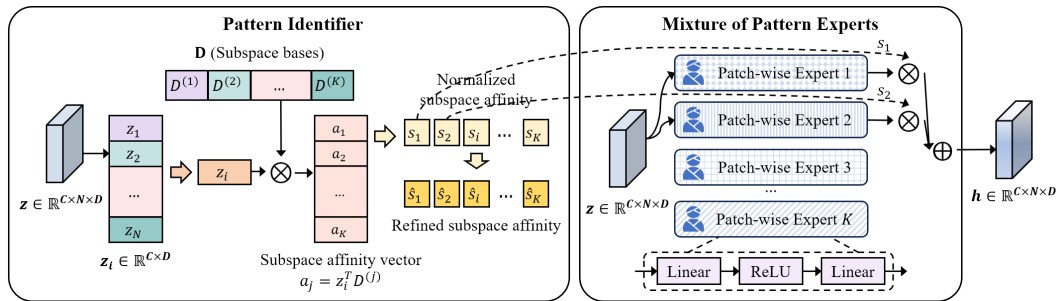

Figure 3: Illustration of the proposed Pattern Identifier and Mixture of Pattern Experts. The embedded representation $\mathbf{z}$ from DDE combines with subspace $\mathbf{D}$ to construct the subspace affinity vector, which yields the normalized subspace affinity $S$. Subsequently, the refined subspace affinity $\hat{S}$ is computed from $S$ to provide self-supervised information. Then, we assign the corresponding patch-wise experts to the embedded representation $\mathbf{z}$ according to $S$ for modeling.

We only keep the real part of the result, and hence, we do not modify the feed-forward layers in the Transformer. The structure of the Frequency Encoder is depicted in Figure 2 (c), yielding frequency features $z_f \in \mathbb{R}^{C \times N \times D}$.

By modeling data in both the time and frequency domains, the DDE provides a more comprehensive understanding of temporal patterns, enabling the model to effectively handle complexities such as concept drift and evolving dynamics. This dual-domain perspective enhances the model's robustness and predictive accuracy, offering a versatile foundation for real-world time series forecasting.

## 3.5 Pattern Identifier

To address the complex and evolving patterns in time series data, we introduce a novel Pattern Identifier (PI) module, an essential innovation within our framework. Unlike traditional approaches that treat the entire time series uniformly, our PI module dynamically classifies patches based on their distributional characteristics, enabling a more precise and adaptive modeling strategy.

The core of our approach lies in leveraging subspace clustering to detect concept shifts across multiple subspaces, as illustrated in Figure 3. The PI module plays a central role by directly analyzing the intrinsic properties of each patch and clustering them into distinct groups based on their latent patterns. In the time domain, PI enables TFPS to identify shifts in temporal characteristics such as seasonality and trends. In the frequency domain, it captures shifts associated with frequency-specific structures, like periodic behaviors and spectral changes, providing a comprehensive perspective on evolving patterns throughout the series.

To provide clarity, Figure 3 showcases an application of the PI module exclusively within the time domain. However, the insights and methodology seamlessly extend to the frequency domain, presenting a unified solution to the challenge of concept shifts.

The PI module iteratively refines subspace bases, which in turn improve representation learning and enable more accurate modeling of evolving patterns. It operates through the following three steps.

**Construction of Subspace Bases.** We define a new variable $\mathbf{D} = [\mathbf{D}^{(1)}, \mathbf{D}^{(2)}, \cdots, \mathbf{D}^{(K)}]$ to represent the bases of $K$ subspaces, where $\mathbf{D}$ consists of $K$ blocks, each $\mathbf{D}^{(j)} \in \mathbf{R}^{q \times d}$, $\left\|\mathbf{D}_u^{(j)}\right\| = 1, u = 1, \cdots d, j = 1, \cdots, K$. To control the column sizes of $\mathbf{D}$, we impose the following constraint:

$$R_1 = \frac{1}{2} \left\|\mathbf{D}^T \mathbf{D} \odot \mathbf{I} - \mathbf{I}\right\|_F^2, \tag{3}$$

where $\odot$ denotes the Hadamard product, and $\mathbf{I}$ is an identity matrix of size $Kd \times Kd$.

**Subspaces Differentiation.** To ensure the dissimilarity between different subspaces, we introduce the second constraint:

$$\begin{aligned} R_2 &= \frac{1}{2}\left\|\mathbf{D}^{(j)T}\mathbf{D}^{(l)}\right\|_F^2, \quad j \neq l, \\ &= \frac{1}{2}\left\|\mathbf{D}^T\mathbf{D} \odot \mathbf{O}\right\|_F^2, \end{aligned} \tag{4}$$

where $\mathbf{O}$ is a matrix with all off-diagonal $d$-size blocks set to 1 and diagonal blocks set to 0. Combining $R_1$ and $R_2$ yields the regularization term for $R$:

$$R = \alpha(R_1 + R_2), \tag{5}$$

where $\alpha$ is a tuning parameters, fixed at $10^{-3}$ in this work.

**Subspace Affinity and Refinement.** We propose a novel subspace affinity measure $\mathbf{S}$ to assess the relationship between the embedded representation $\mathbf{z}$ from DDE and the subspace bases $\mathbf{D}$. The affinity $s_{ij}$, representing the probability that the embedded $\mathbf{z}_i$ belongs to the $j$-th subspace, is defined as:

$$s_{ij} = \frac{\left\|\mathbf{z}_i^T\mathbf{D}^{(j)}\right\|_F^2 + \eta d}{\sum_j(\left\|\mathbf{z}_i^T\mathbf{D}^{(j)}\right\|_F^2 + \eta d)}, \tag{6}$$

where $\eta$ is a parameter controlling the smoothness, fixed to the same value as $d$. To emphasize more confident assignments, we introduce a refined subspace affinity $\hat{s}_{ij}$:

$$\hat{s}_{ij} = \frac{s_{ij}^2/\sum_i s_{ij}}{\sum_j(s_{ij}^2/\sum_i s_{ij})}. \tag{7}$$

This refinement sharpens the clustering by weighting high-confidence assignments more. The subspace clustering objective based on the Kullback-Leibler divergence is:

$$\mathcal{L}_{sub} = KL(\hat{S} \parallel S) = \sum_i \sum_j \hat{s}_{ij} log\frac{\hat{s}_{ij}}{s_{ij}}. \tag{8}$$

The clustering loss is defined as:

$$\mathcal{L}_{PI} = R + \beta\mathcal{L}_{sub}, \tag{9}$$

where $\beta$ is a hyperparameter balancing the regularization and subspace clustering terms. A detailed sensitivity analysis of $\alpha$ and $\beta$ is presented in Appendix J.

### 3.6 Mixture of Pattern Experts

Traditional time series forecasting methods often rely on a uniform distribution modeling (UDM) approach, which struggles to adapt to the complexities of diverse and evolving patterns in real-world data. To address this limitation, we introduce the Mixture of Pattern Experts module (MoPE), which assigns specialized experts to patches based on their unique underlying patterns, enabling more precise and adaptive forecasting.

Given the cluster assignments $s$ obtained from the PI module, we apply the Patch-wise MoPE to the feature tensor $z \in \mathbb{R}^{C \times N \times D}$. The MoPE module consists of the following key components:

**Gating Network.** The gating network $G$ calculates the gating weights for each expert based on the cluster assignment $s$ and selects the top $k$ experts. The gating weights are computed as:

$$G(s) = \text{Softmax}(\text{TopK}(s)). \tag{10}$$

Here, the top $k$ logits are selected and normalized using the Softmax function to produce the gating weights.

**Expert Networks.** The MoPE contains $K$ expert networks, denoted as $E_1, \ldots, E_K$. Each expert network is modeled as an MLP consisting of two linear layers and a ReLU activation. Given a patch-wise feature $z$, each expert network $E_k$ processes the input to generate its respective output.

**Output Aggregation.** The final output $h$ of the MoPE module is a weighted sum of the outputs from all the selected experts, with the weights provided by the gating network:

$$h = \sum_{k=1}^{K} G(s)E_k(z). \tag{11}$$

After the frequency branch is processed by the inverse Fast Fourier transform, the time-frequency outputs $h_t$ and $h_f$, are concatenated to form $h = \text{concat}(h_t, h_f) \in \mathbb{R}^{C \times N \times 2D}$.

Finally, a linear transformation is applied to the disentangled and pattern-specific representations $h$ to generate the prediction: $\hat{Y} = \text{Linear}(h) \in \mathbb{R}^{H \times C}$.

This approach ensures that the MoPE dynamically assigns and aggregates contributions from various experts based on evolving patterns, improving the model's adaptability and accuracy.

### 3.7 Loss Function

Following [44], we use the Mean Squared Error (MSE) loss to quantify the discrepancy between predicted values $\hat{Y}$ and ground truth values $Y$: $\mathcal{L}_{MSE} = (\hat{Y} - Y)^2$. In addition to the MSE loss, we incorporate the clustering regularization loss from the PI module, yielding the final loss function:

$$\mathcal{L} = \mathcal{L}_{MSE} + \mathcal{L}_{PI_t} + \mathcal{L}_{PI_f}. \tag{12}$$

This combined loss ensures that the model not only minimizes forecasting errors but also accurately identifies and maintains the integrity of pattern clusters across time. The algorithm is provided in the Appendix L.

## 4 Experiments

### 4.1 Experimental Setup

**Datasets and Baselines.** We conducted our experiments on nine publicly available real-world multivariate time series datasets, i.e., ETT (ETTh1, ETTh2, ETTm1, ETTm2), Exchange, Weather, Electricity, Traffic, and ILI. These datasets are provided in [62] for time series forecasting. More details about these datasets are included in Appendix A.

We employed a diverse set of state-of-the-art forecasting models as baselines, categorized based on the type of information they utilize as follows. **(1) Time-domain methods:** PatchTST [44], DLinear [69], TimesNet [61] and iTransformer [29]; **(2) Frequency-domain methods:** FEDformer [71] and FITS [64]; **(3) Time-frequency methods:** TFDNet-IK [42] and TSLANet [9]. We rerun all the experiments with codes provided by their official implementation.

In addition, we compare TFPS with recent foundation models, including AutoTimes [30], Moment [16], and Timer [32]. We rerun all experiments for a fair comparison: AutoTimes is reproduced using its official implementation, while Moment and Timer are evaluated based on the OpenLTM [32].

We further include comparisons with normalization techniques, MoE-based architectures, and methods designed to address distribution shifts. Comprehensive results are presented in Appendix G.

**Experiments Details.** Following previous works [44], we used ADAM [21] as the default optimizer across all the experiments. We employed the MSE and mean absolute error (MAE) as the evaluation metrics, where lower values indicate better performance. A detailed explanation is provided in Appendix E. TFPS was implemented by PyTorch [48] and trained on a single NVIDIA RTX 3090 24GB GPU. We conducted grid search to optimize the following three parameters, i.e., learning rate $= \{0.0001, 0.0005, 0.001, 0.005, 0.01, 0.05\}$, the number of experts in the time domain $K_t = \{1, 2, 4, 8\}$, and the number of experts in the frequency domain $K_f = \{1, 2, 4, 8\}$.

### 4.2 Overall Performance Comparison

Table 1 highlights the consistent superiority of TFPS across multiple datasets and prediction horizons, securing the top performance in 57 out of 72 experimental configurations. In particular, TFPS demonstrates significant improvements over time-domain methods, with an overall improvement of

Table 1: Multivariate long-term forecasting results with prediction lengths $H \in \{24, 36, 48, 60\}$ for ILI and $H \in \{96, 192, 336, 720\}$ for others. The input lengths are $L = 104$ for ILI and $L = 96$ for others. The best results are highlighted in **bold** and the second best are underlined.

| Model | | IMP. | TFPS (Our) | | TSLANet (2024) | | FITS (2024) | | iTransformer (2024) | | TFDNet-IK (2023) | | PatchTST (2023) | | TimesNet (2023) | | DLinear (2023) | | FEDformer (2022) | |
|---|---|---|---|---|---|---|---|---|---|---|---|---|---|---|---|---|---|---|---|---|---|
| Metric | | MSE | MSE | MAE | MSE | MAE | MSE | MAE | MSE | MAE | MSE | MAE | MSE | MAE | MSE | MAE | MSE | MAE | MSE | MAE |
| ETTh1 | 96 | -1.1% | 0.398 | 0.413 | 0.387 | 0.405 | 0.395 | **0.403** | 0.387 | 0.405 | 0.396 | 0.409 | 0.413 | 0.419 | 0.389 | 0.412 | 0.398 | 0.410 | **0.385** | 0.425 |
| | 192 | 4.8% | **0.423** | **0.423** | 0.448 | 0.436 | 0.445 | 0.432 | 0.441 | 0.436 | 0.451 | 0.441 | 0.460 | 0.445 | 0.441 | 0.442 | 0.434 | 0.427 | 0.441 | 0.461 |
| | 336 | 1.8% | **0.484** | **0.461** | 0.491 | 0.487 | 0.489 | 0.463 | 0.491 | 0.463 | 0.495 | 0.462 | 0.497 | 0.463 | 0.491 | 0.467 | 0.499 | 0.477 | 0.491 | 0.473 |
| | 720 | 3.0% | **0.488** | **0.476** | 0.505 | 0.486 | 0.496 | 0.485 | 0.509 | 0.494 | 0.492 | 0.482 | 0.501 | 0.486 | 0.512 | 0.491 | 0.508 | 0.503 | 0.501 | 0.499 |
| ETTh2 | 96 | -2.0% | 0.313 | 0.355 | 0.290 | 0.345 | 0.295 | 0.344 | 0.301 | 0.350 | **0.289** | **0.337** | 0.299 | 0.348 | 0.324 | 0.368 | 0.315 | 0.374 | 0.342 | 0.383 |
| | 192 | -2.9% | 0.405 | 0.410 | **0.362** | **0.391** | 0.382 | 0.396 | 0.380 | 0.399 | 0.379 | 0.395 | 0.383 | 0.398 | 0.393 | 0.410 | 0.432 | 0.447 | 0.434 | 0.440 |
| | 336 | 10.5% | **0.392** | **0.415** | 0.401 | 0.419 | 0.416 | 0.425 | 0.424 | 0.432 | 0.416 | 0.422 | 0.424 | 0.431 | 0.429 | 0.437 | 0.486 | 0.481 | 0.512 | 0.497 |
| | 720 | 12.6% | **0.410** | **0.433** | 0.419 | 0.439 | 0.418 | 0.437 | 0.430 | 0.447 | 0.424 | 0.441 | 0.429 | 0.445 | 0.433 | 0.448 | 0.732 | 0.614 | 0.467 | 0.476 |
| ETTm1 | 96 | 4.1% | **0.327** | **0.367** | 0.329 | 0.368 | 0.354 | 0.375 | 0.342 | 0.377 | 0.331 | 0.369 | 0.331 | 0.370 | 0.337 | 0.377 | 0.346 | 0.374 | 0.360 | 0.406 |
| | 192 | 2.6% | **0.374** | 0.395 | 0.376 | 0.383 | 0.392 | 0.393 | 0.383 | 0.396 | 0.376 | **0.381** | 0.374 | 0.395 | 0.395 | 0.406 | 0.382 | 0.392 | 0.395 | 0.427 |
| | 336 | 4.2% | **0.401** | **0.408** | 0.403 | 0.414 | 0.425 | 0.415 | 0.418 | 0.418 | 0.405 | 0.410 | 0.402 | 0.412 | 0.433 | 0.432 | 0.414 | 0.414 | 0.448 | 0.458 |
| | 720 | -0.7% | 0.479 | 0.456 | **0.445** | 0.438 | 0.486 | 0.449 | 0.487 | 0.457 | 0.471 | **0.437** | 0.466 | 0.446 | 0.484 | 0.458 | 0.478 | 0.455 | 0.491 | 0.479 |
| ETTm2 | 96 | 6.9% | **0.170** | **0.255** | 0.179 | 0.261 | 0.183 | 0.266 | 0.186 | 0.272 | 0.176 | 0.267 | 0.177 | 0.260 | 0.182 | 0.262 | 0.184 | 0.276 | 0.193 | 0.285 |
| | 192 | 7.1% | **0.235** | **0.296** | 0.243 | 0.303 | 0.247 | 0.305 | 0.254 | 0.314 | 0.245 | 0.302 | 0.248 | 0.306 | 0.252 | 0.307 | 0.282 | 0.357 | 0.256 | 0.324 |
| | 336 | 4.6% | **0.297** | **0.335** | 0.308 | 0.345 | 0.307 | 0.342 | 0.316 | 0.351 | 0.303 | 0.340 | 0.303 | 0.341 | 0.312 | 0.346 | 0.324 | 0.364 | 0.321 | 0.364 |
| | 720 | 3.6% | **0.401** | **0.397** | 0.403 | 0.400 | 0.407 | 0.401 | 0.414 | 0.407 | 0.405 | 0.399 | 0.405 | 0.403 | 0.417 | 0.404 | 0.441 | 0.454 | 0.434 | 0.426 |
| Exchange | 96 | 12.7% | **0.083** | **0.205** | 0.085 | 0.206 | 0.088 | 0.210 | 0.086 | 0.206 | 0.084 | 0.205 | 0.089 | 0.206 | 0.105 | 0.233 | 0.089 | 0.219 | 0.136 | 0.265 |
| | 192 | 11.2% | **0.174** | **0.297** | 0.178 | 0.300 | 0.181 | 0.304 | 0.181 | 0.304 | 0.176 | 0.299 | 0.178 | 0.302 | 0.219 | 0.342 | 0.180 | 0.319 | 0.279 | 0.384 |
| | 336 | 10.4% | **0.310** | **0.398** | 0.329 | 0.415 | 0.324 | 0.413 | 0.338 | 0.422 | 0.321 | 0.409 | 0.326 | 0.411 | 0.353 | 0.433 | 0.313 | 0.423 | 0.465 | 0.504 |
| | 720 | -13.3% | 1.011 | 0.756 | 0.850 | 0.693 | 0.846 | 0.696 | 0.853 | 0.696 | **0.835** | **0.689** | 0.840 | 0.690 | 0.912 | 0.724 | 0.837 | 0.690 | 1.169 | 0.826 |
| Weather | 96 | 15.6% | **0.154** | **0.202** | 0.176 | 0.216 | 0.167 | 0.214 | 0.176 | 0.216 | 0.165 | 0.216 | 0.177 | 0.219 | 0.168 | 0.218 | 0.197 | 0.257 | 0.236 | 0.325 |
| | 192 | 10.6% | **0.205** | **0.249** | 0.226 | 0.258 | 0.215 | 0.257 | 0.225 | 0.257 | 0.214 | 0.252 | 0.225 | 0.259 | 0.226 | 0.267 | 0.237 | 0.294 | 0.268 | 0.337 |
| | 336 | 9.1% | **0.262** | **0.289** | 0.279 | 0.299 | 0.270 | 0.299 | 0.281 | 0.299 | 0.267 | 0.298 | 0.278 | 0.298 | 0.283 | 0.305 | 0.283 | 0.332 | 0.366 | 0.402 |
| | 720 | 4.1% | **0.344** | **0.342** | 0.355 | 0.355 | 0.347 | 0.345 | 0.358 | 0.350 | 0.347 | 0.346 | 0.351 | 0.346 | 0.355 | 0.353 | 0.347 | 0.382 | 0.407 | 0.422 |
| Electricity | 96 | 14.6% | **0.149** | **0.236** | 0.155 | 0.249 | 0.200 | 0.278 | 0.151 | 0.241 | 0.171 | 0.254 | 0.166 | 0.252 | 0.168 | 0.272 | 0.195 | 0.277 | 0.189 | 0.304 |
| | 192 | 12.0% | **0.162** | **0.253** | 0.170 | 0.264 | 0.200 | 0.281 | 0.167 | 0.258 | 0.189 | 0.269 | 0.174 | 0.261 | 0.186 | 0.289 | 0.194 | 0.281 | 0.198 | 0.312 |
| | 336 | 0.2% | 0.200 | 0.310 | 0.197 | 0.282 | 0.214 | 0.295 | **0.179** | **0.271** | 0.205 | 0.284 | 0.190 | 0.277 | 0.197 | 0.298 | 0.207 | 0.296 | 0.212 | 0.326 |
| | 720 | 7.2% | **0.220** | 0.320 | 0.224 | 0.318 | 0.256 | 0.328 | 0.256 | 0.328 | 0.247 | 0.318 | 0.230 | 0.312 | 0.225 | 0.322 | 0.242 | 0.330 | 0.242 | 0.351 |
| Traffic | 96 | 21.1% | **0.427** | 0.296 | 0.475 | 0.307 | 0.651 | 0.388 | 0.428 | **0.295** | 0.519 | 0.314 | 0.446 | 0.284 | 0.586 | 0.316 | 0.650 | 0.397 | 0.575 | 0.357 |
| | 192 | 17.7% | **0.445** | 0.298 | 0.478 | 0.306 | 0.603 | 0.364 | 0.448 | 0.302 | 0.513 | 0.314 | 0.453 | 0.285 | 0.618 | 0.323 | 0.600 | 0.372 | 0.613 | 0.381 |
| | 336 | 17.0% | **0.459** | 0.307 | 0.494 | 0.312 | 0.610 | 0.366 | 0.465 | 0.311 | 0.525 | 0.319 | 0.467 | 0.291 | 0.634 | 0.337 | 0.606 | 0.374 | 0.622 | 0.380 |
| | 720 | 15.1% | **0.496** | 0.313 | 0.528 | 0.331 | 0.648 | 0.387 | 0.501 | 0.333 | 0.561 | 0.336 | 0.501 | 0.492 | 0.659 | 0.349 | 0.646 | 0.396 | 0.630 | 0.383 |
| ILI | 24 | 40.9% | **1.349** | **0.760** | 1.749 | 0.898 | 3.489 | 1.373 | 2.443 | 1.078 | 1.824 | 0.824 | 1.614 | 0.835 | 1.699 | 0.871 | 2.239 | 1.041 | 3.217 | 1.246 |
| | 36 | 43.6% | **1.239** | **0.752** | 1.754 | 0.912 | 3.530 | 1.370 | 2.455 | 1.086 | 1.699 | 0.813 | 1.475 | 0.859 | 1.733 | 0.913 | 2.238 | 1.049 | 2.688 | 1.074 |
| | 48 | 40.4% | **1.461** | **0.801** | 2.050 | 0.984 | 3.671 | 1.391 | 3.437 | 1.331 | 1.762 | 0.831 | 1.642 | 0.880 | 2.272 | 0.999 | 2.252 | 1.064 | 2.540 | 1.057 |
| | 60 | 39.8% | **1.458** | **0.836** | 2.240 | 1.039 | 4.030 | 1.462 | 2.734 | 1.155 | 1.758 | 0.863 | 1.608 | 0.885 | 1.998 | 0.974 | 2.236 | 1.057 | 2.782 | 1.136 |
| 1st Count | | | 57 | | 3 | | 1 | | 3 | | 6 | | 1 | | 0 | | 0 | | 1 | |

Table 2: Ablation study of TFPS components. The model variants in our ablation study include the following configurations across both time and frequency branches: (a) inclusion of the encoder, PI and MoPE; (b) PI replaced with Linear; (c) only the encoder. The best results are in **bold**.

| Time Branch | | | Frequency Branch | | | ETTh1 | | | | ETTh2 | | | |
|---|---|---|---|---|---|---|---|---|---|---|---|---|---|
| Encoder | PI | MoPE | Encoder | PI | MoPE | 96 | 192 | 336 | 720 | 96 | 192 | 336 | 720 |
| ✓ | ✓ | ✓ | ✓ | ✓ | ✓ | **0.398** | **0.423** | **0.484** | **0.488** | **0.313** | **0.405** | **0.392** | **0.410** |
| ✓ | ✓ | ✓ | | | | 0.401 | 0.459 | 0.486 | 0.492 | 0.318 | 0.409 | 0.400 | 0.428 |
| ✓ | Linear | ✓ | | | | 0.401 | 0.451 | 0.494 | 0.509 | 0.325 | 0.411 | 0.400 | 0.434 |
| ✓ | | | | | | 0.414 | 0.460 | 0.501 | 0.500 | 0.339 | 0.411 | 0.426 | 0.431 |
| | | | ✓ | ✓ | ✓ | 0.455 | 0.507 | 0.539 | 0.576 | 0.324 | 0.407 | 0.417 | 0.436 |
| | | | ✓ | Linear | ✓ | 0.503 | 0.535 | 0.558 | 0.583 | 0.398 | 0.446 | 0.457 | 0.444 |
| | | | ✓ | | | 0.552 | 0.583 | 0.591 | 0.594 | 0.371 | 0.426 | 0.418 | 0.463 |

9.5% in MSE and 6.4% in MAE. Compared to frequency-domain methods, TFPS shows even more pronounced enhancements, with MSE improved by 16.9% and MAE by 12.4%.

While the time-frequency methods like TSLANet and TFDNet perform competitively on several datasets, TFPS still outperforms them, showing improvement of 5.2% in MSE and 2.2% in MAE. These substantial improvements can be attributed to the integration of both time- and frequency-domain information, combined with our innovative approach to modeling distinct patterns with specialized experts. By addressing the underlying concept shifts and capturing complex, evolving patterns in time series data, TFPS achieves more accurate predictions than other baselines.

Table 3: Compared with foundation models.

| Model | IMP. | TFPS (Our) | | AutoTimes (2024) | | Moment (2024) | | Timer (2024) | |
|---|---|---|---|---|---|---|---|---|---|
| Metric | MSE | MSE | MAE | MSE | MAE | MSE | MAE | MSE | MAE |
| ETTh1 | 0.2% | 0.401 | **0.412** | 0.396 | 0.428 | 0.415 | 0.439 | **0.394** | 0.417 |
| ETTh2 | 10.9% | **0.335** | **0.386** | 0.363 | 0.406 | 0.381 | 0.412 | 0.382 | 0.418 |
| ETTm1 | 2.4% | **0.343** | **0.374** | 0.364 | 0.389 | 0.348 | 0.386 | 0.344 | 0.378 |
| ETTm2 | 7.3% | **0.248** | **0.308** | 0.273 | 0.327 | 0.265 | 0.325 | 0.264 | 0.321 |
| Traffic | -3.6% | 0.398 | 0.268 | 0.379 | 0.265 | 0.395 | 0.273 | 0.379 | **0.255** |
| Electricity | 3.7% | **0.159** | **0.249** | 0.168 | 0.261 | 0.163 | 0.263 | 0.165 | 0.258 |

Table 4: Compared with normalization methods.

| Model | IMP. | TFPS | DLinear | | | |
|---|---|---|---|---|---|---|
| | | | SIN | SAN | Dish-TS | RevIN |
| ETTh1 | 1.5% | **0.448** | 0.454 | 0.456 | 0.461 | 0.451 |
| ETTh2 | 2.4% | **0.380** | 0.386 | 0.388 | 0.392 | 0.390 |
| ETTm1 | 2.3% | **0.395** | 0.405 | 0.399 | 0.406 | 0.409 |
| ETTm2 | 3.3% | **0.276** | 0.283 | 0.280 | 0.293 | 0.284 |
| Weather | 5.3% | **0.241** | 0.253 | 0.249 | 0.263 | 0.254 |

## 4.3 Ablation Study

Table 2 presents the MSE results of TFPS and its variants with different combinations of encoders, PI, and MoPE. **1) Best Result.** The full TFPS model, i.e., both the time and frequency branches, along with their respective encoders, PI, and MoPE are included, performs the best across all the forecast horizons for both datasets. **2) Linear vs. PI.** We replace PI with a linear layer and find that it generally results in higher MSE in most cases, indicating that accurately capturing specific patterns is crucial. **3) Impact of Pattern-aware Modeling.** Additionally, when comparing the results with the encoder-only configuration, two variants with MoPE in each branch achieved improved MSE, further supporting the necessity of patter-aware modeling. **4) Importance of DDE.** Furthermore, we find that both the time encoder and frequency encoder alone yield worse performance, with the time encoder playing a more significant role. In summary, incorporating both branches with PI and MoPE provides the best performance, while simpler configurations result in higher MSE. See Appendix K for an in-depth analysis of each component's contribution.

## 4.4 Comparsion with Foundation Models

To ensure a fair comparison with foundation models, we searched input lengths among 96, 192, 336, and 512. The average results across all forecasting lengths are included in Table 3, with detailed results provided in Appendix F.2. As shown in Table 3, TFPS consistently outperforms recent foundation models. Notably, on challenging datasets such as ETTh2 and ETTm2, TFPS achieves substantial improvements in MSE by 10.9% and 7.3%, respectively. Although TFPS performs slightly worse on the Traffic dataset, we attribute this to the relatively mild distribution shift observed in Traffic (see Table 5), which may reduce the benefit of our pattern-specific modeling. These results suggest that TFPS not only matches but often surpasses large-scale foundation models in forecasting accuracy, benefiting from its expert-based design that explicitly captures distributional heterogeneity.

## 4.5 Comparsion with Normalization Methods

Normalization methods can reduce fluctuations to enhance performance and are widely used for non-stationary time series forecasting [17, 34, 10, 31, 20]. We compare our TFPS with these state-of-the-art normalization methods and Table 4 presents the average MSE across all forecasting lengths for each dataset. While normalization improves stability by enforcing distributional consistency, TFPS retains the intrinsic non-stationarity and models diverse patterns through distribution-specific experts, achieving better adaptability and forecasting accuracy. Detailed results are provided in Appendix G.1.

## 4.6 Visualization

We visualize the prediction curves for ETTh1 with $H = 192$. Given that DLinear exhibits competitive performance in Table 1, we compare its results with those of TFPS in Figure 4 under two scenarios: (a) sudden drift caused by external factors or random events, and (b) gradual drift where the trend is dominant. It is evident that DLinear struggles to achieve accurate predictions in both scenarios. In contrast, our TFPS consistently produces accurate forecasts despite these challenges, demonstrating its robustness in dealing with various concept dynamics.

## 4.7 Analysis of Experts

**Qualitative Visualizations of Pattern Identifier.** Through training, pattern experts in MoPE spontaneously specialize, and we present two examples in Figure 5. We visualize the expert with the highest score as the routed expert for each instance pair. In the provided examples, we observe that

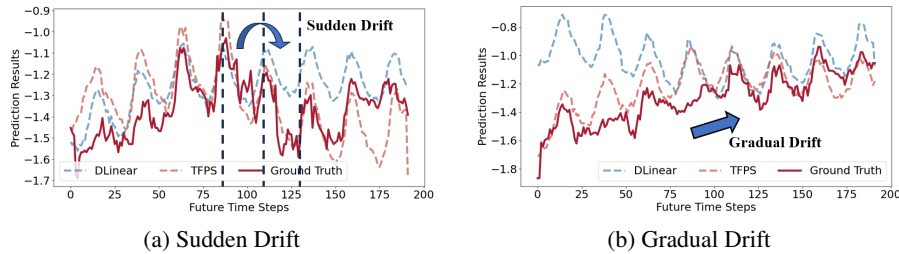

(a) Sudden Drift        (b) Gradual Drift

Figure 4: Visualizations of DLinear and TFPS on the ETTh1 dataset when $H = 192$.

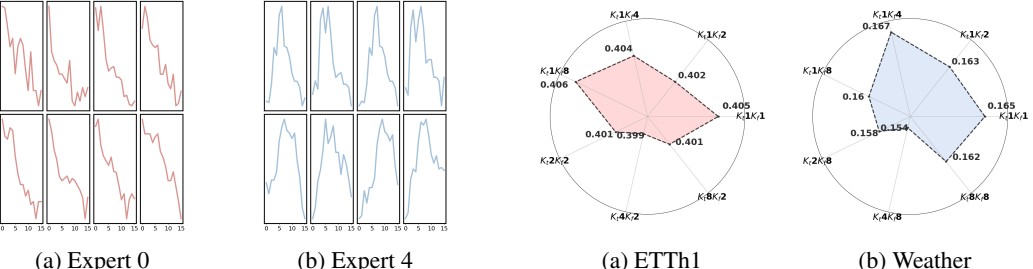

(a) Expert 0     (b) Expert 4        (a) ETTh1     (b) Weather

Figure 5: Interpretable patterns via PI. Expert-0 specializes in downward trends, while Expert-4 focuses on parabolic trends.

Figure 6: Experiments on the number of experts when $H = 96$. Further analysis of expert behavior is provided in Appendix H.

expert-0 specialize in downward-related concepts, while expert-4 focuses on parabolic trend. These examples also demonstrate the interpretability of MoPE.

**Number of Experts.** In Figure 6, we set the learning rate to 0.0001 and conducted four sets of experiments on the ETTh1 and Weather datasets, $K_t = 1$, $K_f = \{1, 2, 4, 8\}$, to explore the effect of the number of frequency experts on the results. For example, $K_t 1 K_f 4$ means that the TFPS contains 1 time experts and 4 frequency experts. We observed that $K_t 1 K_f 2$ outperformed $K_t 1 K_f 4$ in both cases, suggesting that increasing the number of experts does not always lead to better performance.

In addition, we conducted three experiments based on the optimal number of frequency experts to verify the impact of varying the number of time experts on the results. As shown in Figure 6, the best results for ETTh1 were obtained with $K_t 4 K_f 2$, while for Weather, the optimal results were achieved with $K_t 4 K_f 8$. Combined with the average Wasserstein distance in Table 5, we attribute this to the fact that, in cases where concept drift is more severe, such as Weather, more experts are needed, whereas fewer experts are sufficient when the drift is less severe.

## 5 Conclusion

In this paper, we propose a novel pattern-aware time series forecasting framework, TFPS, which incorporates a dual-domain mixture of pattern experts approach. Our TFPS framework aims to address the distribution shift across time series patches and effectively assigns pattern-specific experts to model them. Experimental results across eight diverse datasets demonstrate that TFPS surpasses state-of-the-art methods in both quantitative metrics and visualizations. Future work will focus on investigating evolving distribution shifts, particularly those introduced by the emergence of new patterns, such as unforeseen epidemics or outbreaks.

## 6 Acknowledgments and Disclosure of Funding

This work was supported in part by the National Natural Science Foundation of China under Grants U23B2049, 62376194, 61925602, 62406219, and 62436001; in part by the China Postdoctoral Science Foundation - Tianjin Joint Support Program under Grant 2023T014TJ; and in part by the China Scholarship Council under Grant 202406250137.

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

# A  Dataset

We evaluate the performance of TFPS on eight widely used datasets, including four ETT datasets (ETTh1, ETTh2, ETTm1 and ETTm2), Exchange, Weather, Electricity, and ILI. This subsection provides a summary of the datasets:

- **ETT** [2] [70] (Electricity Transformer Temperature) dataset contains two electric transformers, ETT1 and ETT2, collected from two separate counties. Each of them has two versions of sampling resolutions (15min & 1h). Thus, there are four ETT datasets: **ETTm1**, **ETTm2**, **ETTh1**, and **ETTh2**.
- **Exchange-Rate** [3] [23] the exchange-rate dataset contains the daily exchange rates of eight foreign countries including Australia, British, Canada, Switzerland, China, Japan, New Zealand, and Singapore ranging from 1990 to 2016.
- **Weather** [4] [62] dataset contains 21 meteorological indicators in Germany, such as humidity and air temperature.
- **Electricity** [5] [62] is a dataset that describes 321 customers' hourly electricity consumption.
- **Traffic** [6] [62] is a dataset featuring hourly road occupancy rates from 862 sensors along the freeways in the San Francisco Bay area.
- **ILI** [7] [62] dataset collects the number of patients and influenza-like illness ratio in a weekly frequency.

For the data split, we follow [69] and split the data into training, validation, and testing by a ratio of 6:2:2 for the ETT datasets and 7:1:2 for the others. Details are shown in Table 5. The best parameters are selected based on the lowest validation loss and then applied to the test set for performance evaluation. The data and codes are available: `https://github.com/syrGitHub/TFPS`.

Table 5: The statistics of the datasets.

| Datasets | Variates | Prediction Length | Timesteps | Granularity | Average Wasserstein[*] (Time Domain) | Average Wasserstein[*] (Frequency Domain) |
|---|---|---|---|---|---|---|
| ETTh1 | 7 | {96, 192, 336, 720} | 17,420 | 1 hour | 9.268 | 11.561 |
| ETTh2 | 7 | {96, 192, 336, 720} | 17,420 | 1 hour | 13.221 | 18.970 |
| ETTm1 | 7 | {96, 192, 336, 720} | 69,680 | 15 min | 9.336 | 10.660 |
| ETTm2 | 7 | {96, 192, 336, 720} | 69,680 | 15 min | 13.606 | 16.574 |
| Exchange-Rate | 8 | {96, 192, 336, 720} | 7,588 | 1 day | 0.132 | 0.144 |
| Weather | 21 | {96, 192, 336, 720} | 52,696 | 10 min | 39.742 | 77.422 |
| Electricity | 321 | {96, 192, 336, 720} | 26,304 | 1 hour | 520.162 | 1018.311 |
| Traffic | 862 | {96, 192, 336, 720} | 17,451 | 1 hour | 0.011 | 0.028 |
| ILI | 7 | {24, 36, 48, 60} | 966 | 1 week | 258881.714 | 381377.494 |

[*] A large Wasserstein indicates a more severe drift.

# B  Related Work

Deep learning has achieved remarkable success across diverse domains such as computer vision [14, 38, 39, 40, 37], natural language processing [58, 6], and multi-modality [13, 15], and has also advanced the state of the art in time series modeling [56, 19].

**The Combination of Time and Frequency Domains.** Time-domain models excel at capturing sequential trends, while frequency-domain models are essential for identifying periodic and oscillatory patterns. Recent research has increasingly emphasized integrating information from both domains to better interpret underlying patterns. For instance, ATFN [66] demonstrates the advantage of frequency domain methods for forecasting strongly periodic time series through a time–frequency adaptive

---

[2] `https://github.com/zhouhaoyi/ETDataset`

[3] `https://github.com/laiguokun/multivariate-time-series-data`

[4] `https://www.bgc-jena.mpg.de/wetter/`

[5] `https://archive.ics.uci.edu/ml/datasets/ElectricityLoadDiagrams20112014`

[6] `http://pems.dot.ca.gov`

[7] `https://gis.cdc.gov/grasp/fluview/fluportaldashboard.html`

network. TFDNet [42] adopts a branching structure to capture long-term latent patterns and temporal periodicity from both domains. Similarly, JTFT [4] utilizes the frequency domain representation to extract multi-scale dependencies while enhancing local relationships modeling through time domain representation. TFMRN [65] expands data in both domains to capture finer details that may not be evident in the original data. Recently, TSLANet [9] leverages Fourier analysis to enhance feature representation and capture both long-term and short-term interactions.

Building on these approaches, our proposed method, TFPS, introduces a novel Dual-Domain Encoder that effectively combines time and frequency domain information to capture both trend and periodic patterns. By integrating time-frequency features, TFPS significantly advances the field in addressing the complexities inherent in time series forecasting.

**Mixture-of-Experts.** Mixture-of-Experts (MoE) models have gained attention for their ability to scale efficiently by activating only a subset of experts for each input, as first introduced by [57]. Despite their success, challenges such as training instability, expert redundancy, and limited expert specialization have been identified [50, 5]. These issues hinder the full potential of MoE models in real-world tasks.

Recent advances have integrated MoE with Transformers to improve scalability and efficiency. For example, GLaM [7] and Switch Transformer [12] interleave MoE layers with Transformer blocks, reducing computational costs. Other models like state space models (SSMs) [49, 2], [1] combines MoE with alternative architectures for enhanced scalability and inference speed.

In contrast, our approach introduces MoE into time series forecasting by assigning experts to specific time-frequency patterns, enabling more effective, patch-level adaptation. This approach represents a significant innovation in time series forecasting, offering a more targeted and effective way to handle varying patterns across both time and frequency domains.

## C   Wasserstein Distance

The Wasserstein distance, also called the Earth mover's distance or the optimal transport distance, is a similarity metric between two probability distributions. In the discrete case, the Wasserstein distance can be understood as the cost of an optimal transport plan to convert one distribution into the other. The cost is calculated as the product of the amount of probability mass being moved and the distance it is being moved.

Given two one-dimensional probability mass functions, $u$ and $v$, the first Wasserstein distance between them is defined as:

$$l_1(u, v) = \inf_{\pi \in \Gamma(u,v)} \int_{\mathbb{R} \times \mathbb{R}} |x - y| \, d\pi(x, y), \tag{13}$$

where $\Gamma(u, v)$ is the set of (probability) distributions on $\mathbb{R} \times \mathbb{R}$ whose marginals are $u$ and $v$ on the first and second factors respectively. Here, $u(x)$ represents the probability of $u$ at position $x$, with the same interpretation for $v(x)$.

In the special case of one-dimensional distributions, the Wasserstein distance can be equivalently expressed using their cumulative distribution functions (CDFs), $U$ and $V$, as:

$$l_1(u, v) = \int_{-\infty}^{+\infty} |U - V|. \tag{14}$$

This equivalence is rigorously proved in [54].

The input distributions can be empirical, therefore coming from samples whose values are effectively inputs of the function, or they can be seen as generalized functions, in which case they are weighted sums of Dirac delta functions located at the specified values.

## D   Distribution Shifts in both Time and Frequency Domains

The time series $\mathcal{X}$ is segmented into $N$ patches, where each patch $\mathcal{P}_n = \{x_{n1}, x_{n2}, \ldots, x_{nP}\}$ consists of $P$ consecutive timesteps for $n = 1, 2, \cdots, N$. For the frequency domain, we apply a Fourier transform $\mathcal{F}$ to each patch $\mathcal{P}_n$, obtaining its frequency-domain representation as $\hat{\mathcal{P}}_n = \mathcal{F}(\mathcal{P}_n)$.

Each patch's probability distribution in the time domain is denoted as $p_t(\mathcal{P}_n)$, representing the statistical properties of $\mathcal{P}_n$, while its frequency domain distribution, denoted as $p_f(\hat{\mathcal{P}}_n)$, captures its spectral characteristics.

The distribution shifts between two patches $\mathcal{P}_i$ and $\mathcal{P}_j$ are characterized by the comparing their probability distributions in both time and frequency domains. These shifts are defined as:

$$\mathcal{D}_t(\mathcal{P}_i, \mathcal{P}_j) = |d(p_t(\mathcal{P}_i), p_t(\mathcal{P}_j))| > \theta, \tag{15}$$

$$\mathcal{D}_f(\hat{\mathcal{P}}_i, \hat{\mathcal{P}}_j) = |d(p_f(\hat{\mathcal{P}}_i), p_f(\hat{\mathcal{P}}_j))| > \theta, \tag{16}$$

where $d$ is a distance metric, such as Wasserstein distance or Kullback-Leibler divergence, and $\theta$ is a threshold indicating a significant distribution shift. If $\mathcal{D}_t(\mathcal{P}_i, \mathcal{P}_j)$ or $\mathcal{D}_f(\hat{\mathcal{P}}_i, \hat{\mathcal{P}}_j)$ exceeds $\theta$, this implies a significant distribution shift between the two patches in either domain. It is important to note that $\theta$ serves only as a conceptual threshold for defining distribution shifts in the analysis and does not participate in the modeling or training process of TFPS.

## E    Metric Illustration

We use mean square error (MSE) and mean absolute error (MAE) as our metrics for evaluation of all forecasting models. Then calculation of MSE and MAE can be described as:

$$\text{MSE} = \frac{1}{H} \sum_{i=L+1}^{L+H} (\hat{Y}_i - Y_i)^2, \tag{17}$$

$$\text{MAE} = \frac{1}{H} \sum_{i=L+1}^{L+H} \left| \hat{Y}_i - Y_i \right|, \tag{18}$$

where $\hat{Y}$ is predicted vector with $H$ future values, while $Y$ is the ground truth.

In addition, we report the IMP (Improvement) metric, which is defined as:

$$\text{IMP} = \frac{\text{Avg MSE of baselines} - \text{MSE of TFPS}}{\text{Avg MSE of baselines}} \times 100\%. \tag{19}$$

This metric quantifies the relative percentage improvement of TFPS over the average MSE of all baseline methods. A higher IMP value indicates better overall performance of TFPS compared to the baselines.

## F    Hyperparameter-search Results

### F.1    Comparsion with Specific Models

To ensure a fair comparison between models, we conducted experiments using unified parameters and reported results in the main text.

In addition, considering that the reported results in different papers are mostly obtained through hyperparameter search, we provide the experiment results with the full version of the parameter search. We searched for input length among 96, 192, 336, and 512. The results are included in Table 6. All baselines are reproduced by their official code.

We can find that the relative promotion of TFPS over TFDNet is smaller under comprehensive hyperparameter search than the unified hyperparameter setting. It is worth noticing that TFPS runs much faster than TFDNet according to the efficiency comparison in Table 12. Therefore, considering performance, hyperparameter-search cost and efficiency, we believe TFPS is a practical model in real-world applications and is valuable to deep time series forecasting community.

### F.2    Comparsion with Foundation Models

Table 7 presents the detailed results of TFPS compared to recent foundation models (AutoTimes [30], Moment [16], and Timer [32]) across six datasets and four forecasting lengths. Consistent

Table 6: Experiment results under hyperparameter searching for the long-term forecasting task. The best results are highlighted in **bold** and the second best are underlined.

| Model | | IMP. | TFPS (Our) | | TSLANet (2024) | | FITS (2024) | | iTransformer (2024) | | TFDNet-IK (2023) | | PatchTST (2023) | | TimesNet (2023) | | Dlinear (2023) | | FEDformer (2022) | |
|---|---|---|---|---|---|---|---|---|---|---|---|---|---|---|---|---|---|---|---|---|---|
| Metric | | MSE | MSE | MAE | MSE | MAE | MSE | MAE | MSE | MAE | MSE | MAE | MSE | MAE | MSE | MAE | MSE | MAE | MSE | MAE |
| ETTh1 | 96 | 1.5% | 0.372 | 0.404 | 0.368 | 0.394 | 0.374 | 0.395 | 0.387 | 0.405 | **0.360** | **0.387** | 0.375 | 0.400 | 0.389 | 0.412 | 0.384 | 0.405 | 0.385 | 0.425 |
| | 192 | 5.7% | **0.401** | **0.410** | 0.413 | 0.418 | 0.407 | 0.414 | 0.441 | 0.436 | 0.403 | 0.412 | 0.414 | 0.421 | 0.441 | 0.442 | 0.443 | 0.450 | 0.441 | 0.461 |
| | 336 | 9.8% | **0.409** | **0.402** | 0.412 | 0.416 | 0.429 | 0.428 | 0.491 | 0.463 | 0.434 | 0.452 | 0.432 | 0.436 | 0.450 | 0.466 | 0.447 | 0.448 | 0.491 | 0.473 |
| | 720 | 11.2% | **0.423** | **0.433** | 0.473 | 0.477 | 0.425 | 0.446 | 0.509 | 0.494 | 0.437 | 0.452 | 0.450 | 0.466 | 0.512 | 0.491 | 0.504 | 0.515 | 0.501 | 0.499 |
| ETTh2 | 96 | 9.3% | **0.268** | **0.325** | 0.283 | 0.344 | 0.274 | 0.337 | 0.301 | 0.350 | 0.271 | 0.329 | 0.278 | 0.336 | 0.324 | 0.368 | 0.290 | 0.353 | 0.342 | 0.383 |
| | 192 | 10.4% | **0.329** | 0.376 | 0.331 | 0.378 | 0.337 | 0.377 | 0.380 | 0.399 | 0.333 | **0.372** | 0.339 | 0.380 | 0.393 | 0.410 | 0.388 | 0.422 | 0.434 | 0.440 |
| | 336 | 17.7% | 0.329 | 0.401 | **0.319** | **0.377** | 0.360 | 0.398 | 0.424 | 0.432 | 0.361 | 0.396 | 0.336 | 0.380 | 0.429 | 0.437 | 0.463 | 0.473 | 0.512 | 0.497 |
| | 720 | 9.0% | 0.412 | 0.441 | 0.407 | 0.449 | 0.386 | 0.423 | 0.430 | 0.447 | 0.382 | **0.418** | **0.382** | 0.421 | 0.433 | 0.448 | 0.733 | 0.606 | 0.467 | 0.476 |
| ETTm1 | 96 | 10.2% | **0.281** | **0.329** | 0.291 | 0.353 | 0.303 | 0.345 | 0.342 | 0.377 | 0.283 | 0.330 | 0.288 | 0.342 | 0.337 | 0.377 | 0.301 | 0.345 | 0.360 | 0.406 |
| | 192 | 8.5% | **0.324** | **0.354** | 0.329 | 0.372 | 0.337 | 0.365 | 0.383 | 0.396 | 0.327 | 0.356 | 0.334 | 0.372 | 0.395 | 0.406 | 0.336 | 0.366 | 0.395 | 0.427 |
| | 336 | 8.2% | 0.359 | 0.404 | **0.357** | 0.392 | 0.372 | 0.385 | 0.418 | 0.418 | 0.361 | **0.375** | 0.367 | 0.393 | 0.433 | 0.432 | 0.372 | 0.389 | 0.448 | 0.458 |
| | 720 | 8.2% | **0.409** | **0.408** | 0.423 | 0.425 | 0.428 | 0.416 | 0.487 | 0.457 | 0.411 | 0.409 | 0.417 | 0.422 | 0.484 | 0.458 | 0.427 | 0.423 | 0.491 | 0.479 |
| ETTm2 | 96 | 8.9% | **0.158** | **0.243** | 0.167 | 0.256 | 0.165 | 0.255 | 0.186 | 0.272 | 0.158 | 0.244 | 0.164 | 0.253 | 0.182 | 0.262 | 0.172 | 0.267 | 0.193 | 0.285 |
| | 192 | 5.7% | 0.222 | 0.302 | 0.221 | 0.294 | 0.220 | 0.291 | 0.254 | 0.314 | **0.219** | **0.282** | 0.221 | 0.292 | 0.252 | 0.307 | 0.237 | 0.314 | 0.256 | 0.324 |
| | 336 | 8.5% | **0.268** | **0.316** | 0.277 | 0.329 | 0.274 | 0.326 | 0.316 | 0.351 | 0.273 | 0.317 | 0.277 | 0.329 | 0.312 | 0.346 | 0.295 | 0.359 | 0.321 | 0.364 |
| | 720 | 12.0% | **0.344** | **0.373** | 0.356 | 0.382 | 0.367 | 0.383 | 0.414 | 0.407 | 0.346 | 0.374 | 0.365 | 0.384 | 0.417 | 0.404 | 0.427 | 0.439 | 0.434 | 0.426 |
| Traffic | 96 | 17.8% | **0.370** | **0.250** | 0.375 | 0.260 | 0.398 | 0.285 | 0.428 | 0.295 | 0.377 | 0.253 | 0.373 | 0.259 | 0.586 | 0.316 | 0.413 | 0.287 | 0.575 | 0.357 |
| | 192 | 17.0% | **0.390** | **0.258** | 0.395 | 0.272 | 0.408 | 0.288 | 0.448 | 0.302 | 0.391 | 0.260 | 0.395 | 0.273 | 0.618 | 0.323 | 0.424 | 0.290 | 0.613 | 0.381 |
| | 336 | 17.2% | **0.401** | 0.271 | 0.402 | 0.272 | 0.420 | 0.292 | 0.465 | 0.311 | 0.408 | **0.266** | 0.402 | 0.274 | 0.634 | 0.337 | 0.438 | 0.299 | 0.622 | 0.380 |
| | 720 | 15.7% | 0.432 | 0.294 | **0.431** | **0.288** | 0.448 | 0.310 | 0.501 | 0.333 | 0.451 | 0.291 | 0.435 | 0.293 | 0.659 | 0.349 | 0.466 | 0.316 | 0.630 | 0.383 |
| Electricity | 96 | 10.3% | **0.128** | **0.220** | 0.137 | 0.229 | 0.135 | 0.231 | 0.148 | 0.239 | 0.130 | 0.222 | 0.130 | 0.223 | 0.168 | 0.272 | 0.140 | 0.237 | 0.188 | 0.303 |
| | 192 | 11.9% | **0.145** | **0.235** | 0.153 | 0.242 | 0.149 | 0.244 | 0.167 | 0.258 | 0.146 | 0.237 | 0.149 | 0.240 | 0.186 | 0.289 | 0.154 | 0.250 | 0.197 | 0.311 |
| | 336 | 6.8% | 0.166 | 0.258 | 0.165 | 0.263 | 0.165 | 0.260 | 0.178 | 0.271 | **0.162** | **0.254** | 0.168 | 0.262 | 0.196 | 0.297 | 0.169 | 0.268 | 0.212 | 0.327 |
| | 720 | 6.9% | **0.198** | **0.283** | 0.206 | 0.294 | 0.204 | 0.293 | 0.211 | 0.300 | 0.201 | 0.287 | 0.204 | 0.289 | 0.235 | 0.329 | 0.204 | 0.300 | 0.243 | 0.352 |
| 1st Count | | | 32 | | 5 | | 0 | | 0 | | 10 | | 1 | | 0 | | 0 | | 0 | |

Table 7: Detailed results of the comparison between TFPS and foundation models. The best results are highlighted in **bold** and the second best are underlined.

| Model | | IMP. | TFPS (Our) | | AutoTimes (2024) | | Moment (2024) | | Timer (2024) | |
|---|---|---|---|---|---|---|---|---|---|---|
| Metric | | MSE | MSE | MAE | MSE | MAE | MSE | MAE | MSE | MAE |
| ETTh1 | 96 | -2.2% | 0.372 | 0.404 | 0.365 | 0.405 | 0.369 | 0.406 | **0.359** | **0.392** |
| | 192 | -1.3% | 0.401 | **0.410** | 0.392 | 0.423 | 0.405 | 0.431 | **0.391** | 0.413 |
| | 336 | 0.5% | 0.409 | **0.402** | **0.406** | 0.433 | 0.420 | 0.441 | 0.407 | 0.424 |
| | 720 | 3.1% | 0.423 | **0.433** | 0.423 | 0.450 | 0.466 | 0.479 | **0.421** | 0.441 |
| ETTh2 | 96 | 6.3% | **0.268** | **0.325** | 0.286 | 0.349 | 0.287 | 0.347 | 0.285 | 0.344 |
| | 192 | 9.2% | **0.329** | **0.376** | 0.351 | 0.393 | 0.371 | 0.401 | 0.365 | 0.400 |
| | 336 | 17.2% | **0.329** | **0.401** | 0.377 | 0.417 | 0.404 | 0.425 | 0.412 | 0.440 |
| | 720 | 9.8% | **0.412** | **0.441** | 0.439 | 0.464 | 0.463 | 0.476 | 0.467 | 0.487 |
| ETTm1 | 96 | 1.3% | 0.281 | **0.329** | 0.297 | 0.350 | 0.281 | 0.343 | **0.276** | 0.335 |
| | 192 | 1.2% | 0.324 | **0.354** | 0.344 | 0.377 | **0.318** | 0.368 | 0.323 | 0.365 |
| | 336 | 1.6% | 0.359 | 0.404 | 0.380 | 0.398 | **0.356** | 0.391 | 0.358 | **0.388** |
| | 720 | 4.8% | **0.409** | **0.408** | 0.433 | 0.431 | 0.438 | 0.441 | 0.419 | 0.423 |
| ETTm2 | 96 | 8.6% | **0.158** | **0.243** | 0.181 | 0.266 | 0.170 | 0.258 | 0.167 | 0.254 |
| | 192 | 5.5% | **0.222** | 0.302 | 0.241 | 0.306 | 0.233 | 0.301 | 0.229 | **0.297** |
| | 336 | 6.9% | **0.268** | **0.316** | 0.295 | 0.341 | 0.287 | 0.340 | 0.282 | 0.335 |
| | 720 | 8.1% | **0.344** | **0.373** | 0.376 | 0.393 | 0.371 | 0.399 | 0.376 | 0.398 |
| Traffic | 96 | -5.5% | 0.370 | 0.250 | 0.347 | 0.249 | 0.360 | 0.254 | **0.345** | **0.237** |
| | 192 | -5.3% | 0.390 | 0.258 | 0.366 | 0.258 | 0.381 | 0.265 | **0.365** | **0.247** |
| | 336 | -3.1% | 0.401 | 0.271 | 0.383 | 0.267 | 0.404 | 0.277 | **0.381** | **0.256** |
| | 720 | -1.1% | 0.432 | 0.294 | **0.420** | 0.286 | 0.438 | 0.297 | 0.424 | **0.280** |
| Electricity | 96 | 3.4% | **0.128** | **0.220** | 0.135 | 0.230 | 0.133 | 0.236 | 0.130 | 0.224 |
| | 192 | 4.3% | **0.145** | **0.235** | 0.153 | 0.247 | 0.152 | 0.254 | 0.149 | 0.243 |
| | 336 | 1.5% | **0.166** | **0.258** | 0.172 | 0.266 | 0.166 | 0.265 | 0.168 | 0.263 |
| | 720 | 5.2% | **0.198** | **0.283** | 0.212 | 0.300 | 0.202 | 0.298 | 0.213 | 0.303 |
| 1st Count | | | 30 | | 2 | | 2 | | 14 | |

with Table 6, for TFPS, we searched input lengths among 96, 192, 336, and 512, and report the best-performing configuration for each forecasting length.

TFPS achieves the best performance in **30 out of 48** settings, significantly outperforming AutoTimes and Moment, which only achieve 2 best scores each. Although Timer demonstrates competitive

Table 8: Detailed results of the comparison between TFPS and normalization-based methods using FEDformer. The best results are highlighted in **bold** and the second best are underlined.

| Model | | IMP. | TFPS (Our) | | FEDformer | | | | | | | |
|---|---|---|---|---|---|---|---|---|---|---|---|---|
| | | | | | +SIN (2024) | | +SAN (2023) | | +Dish-TS (2023) | | +RevIN (2021) | |
| Metric | | MSE | MSE | MAE | MSE | MAE | MSE | MAE | MSE | MAE | MSE | MAE |
| ETTh1 | 96 | -0.9% | 0.398 | 0.413 | 0.413 | **0.372** | **0.383** | 0.409 | 0.390 | 0.424 | 0.392 | 0.413 |
| | 192 | 3.7% | **0.423** | 0.423 | 0.443 | **0.417** | 0.431 | 0.438 | 0.441 | 0.458 | 0.443 | 0.444 |
| | 336 | -0.5% | 0.484 | 0.461 | **0.465** | **0.448** | 0.471 | 0.456 | 0.495 | 0.486 | 0.495 | 0.467 |
| | 720 | 4.8% | **0.488** | **0.476** | 0.509 | 0.490 | 0.504 | 0.488 | 0.519 | 0.509 | 0.520 | 0.498 |
| ETTh2 | 96 | 34.0% | 0.313 | **0.355** | 0.412 | 0.357 | **0.300** | 0.355 | 0.806 | 0.589 | 0.380 | 0.402 |
| | 192 | 28.3% | 0.405 | **0.410** | 0.472 | 0.453 | **0.392** | 0.413 | 0.936 | 0.659 | 0.457 | 0.443 |
| | 336 | 38.2% | **0.392** | **0.415** | 0.527 | 0.527 | 0.459 | 0.462 | 1.039 | 0.702 | 0.515 | 0.479 |
| | 720 | 41.4% | **0.410** | **0.433** | 0.593 | 0.639 | 0.462 | 0.472 | 1.237 | 0.759 | 0.507 | 0.487 |
| ETTm1 | 96 | 4.5% | 0.327 | 0.367 | 0.373 | **0.320** | **0.311** | 0.355 | 0.348 | 0.397 | 0.340 | 0.385 |
| | 192 | 2.9% | 0.374 | 0.395 | 0.394 | **0.366** | **0.351** | 0.383 | 0.406 | 0.428 | 0.390 | 0.411 |
| | 336 | 4.4% | 0.401 | 0.408 | 0.418 | **0.405** | **0.390** | 0.407 | 0.438 | 0.450 | 0.432 | 0.436 |
| | 720 | -0.8% | 0.479 | 0.456 | **0.451** | 0.475 | 0.456 | **0.444** | 0.497 | 0.481 | 0.497 | 0.466 |
| ETTm2 | 96 | 37.5% | **0.170** | 0.255 | 0.326 | **0.211** | 0.175 | 0.266 | 0.394 | 0.395 | 0.192 | 0.272 |
| | 192 | 35.9% | **0.235** | **0.296** | 0.402 | 0.316 | 0.246 | 0.315 | 0.552 | 0.472 | 0.270 | 0.320 |
| | 336 | 38.6% | **0.297** | **0.335** | 0.465 | 0.399 | 0.315 | 0.362 | 0.808 | 0.601 | 0.348 | 0.367 |
| | 720 | 40.2% | **0.401** | **0.397** | 0.555 | 0.547 | 0.412 | 0.422 | 1.282 | 0.771 | 0.430 | 0.415 |
| Weather | 96 | 30.7% | **0.154** | **0.202** | 0.280 | 0.215 | 0.179 | 0.239 | 0.244 | 0.317 | 0.187 | 0.234 |
| | 192 | 25.6% | **0.205** | **0.249** | 0.314 | 0.264 | 0.234 | 0.296 | 0.320 | 0.380 | 0.235 | 0.272 |
| | 336 | 22.0% | **0.262** | **0.289** | 0.329 | 0.293 | 0.304 | 0.348 | 0.424 | 0.452 | 0.287 | 0.307 |
| | 720 | 21.3% | **0.344** | **0.342** | 0.382 | 0.370 | 0.400 | 0.404 | 0.604 | 0.553 | 0.361 | 0.353 |
| 1st Count | | | 24 | | 9 | | 7 | | 0 | | 0 | |

Table 9: Detailed results of the comparison between TFPS and normalization-based methods using DLinear. The best results are highlighted in **bold** and the second best are underlined.

| Model | | IMP. | TFPS (Our) | | DLinear | | | | | | | |
|---|---|---|---|---|---|---|---|---|---|---|---|---|
| | | | | | +SIN (2024) | | +SAN (2023) | | +Dish-TS (2023) | | +RevIN (2021) | |
| Metric | | MSE | MSE | MAE | MSE | MAE | MSE | MAE | MSE | MAE | MSE | MAE |
| ETTh1 | 96 | -1.6% | 0.398 | 0.413 | 0.401 | 0.415 | **0.385** | **0.395** | 0.389 | 0.399 | 0.393 | 0.416 |
| | 192 | 2.9% | **0.423** | **0.423** | 0.438 | 0.456 | 0.432 | 0.423 | 0.443 | 0.433 | 0.431 | 0.428 |
| | 336 | -0.4% | 0.484 | 0.461 | **0.462** | **0.446** | 0.490 | 0.463 | 0.487 | 0.456 | 0.488 | 0.483 |
| | 720 | 4.6% | **0.488** | **0.476** | 0.515 | 0.500 | 0.516 | 0.504 | 0.523 | 0.508 | 0.493 | 0.482 |
| ETTh2 | 96 | 4.9% | **0.313** | **0.355** | 0.359 | 0.359 | 0.319 | 0.364 | 0.317 | 0.365 | 0.322 | 0.361 |
| | 192 | 1.1% | **0.405** | **0.410** | 0.409 | 0.424 | 0.407 | 0.439 | 0.408 | 0.420 | 0.412 | 0.424 |
| | 336 | 3.6% | **0.392** | **0.415** | 0.398 | 0.429 | 0.411 | 0.425 | 0.416 | 0.426 | 0.403 | 0.427 |
| | 720 | 2.7% | **0.410** | **0.433** | 0.419 | 0.442 | 0.417 | 0.441 | 0.428 | 0.439 | 0.422 | 0.446 |
| ETTm1 | 96 | 4.9% | **0.327** | **0.367** | 0.350 | 0.383 | 0.333 | 0.374 | 0.343 | 0.375 | 0.352 | 0.369 |
| | 192 | 1.9% | **0.374** | 0.395 | 0.383 | 0.396 | 0.374 | 0.396 | 0.381 | **0.391** | 0.388 | 0.396 |
| | 336 | 3.0% | **0.401** | **0.408** | 0.413 | 0.416 | 0.406 | 0.418 | 0.416 | 0.417 | 0.419 | 0.414 |
| | 720 | 0.0% | 0.479 | 0.456 | **0.473** | 0.452 | 0.483 | **0.451** | 0.482 | 0.458 | 0.478 | 0.463 |
| ETTm2 | 96 | 5.5% | **0.170** | **0.255** | 0.172 | 0.283 | 0.179 | 0.272 | 0.189 | 0.264 | 0.179 | 0.269 |
| | 192 | 4.4% | **0.235** | **0.296** | 0.249 | 0.301 | 0.239 | 0.316 | 0.249 | 0.302 | 0.248 | 0.302 |
| | 336 | 1.2% | **0.297** | **0.335** | 0.299 | 0.339 | 0.301 | 0.353 | 0.305 | 0.349 | 0.299 | 0.345 |
| | 720 | 3.2% | **0.401** | **0.397** | 0.412 | 0.421 | 0.404 | 0.408 | 0.429 | 0.402 | 0.411 | 0.402 |
| Weather | 96 | 4.5% | **0.154** | **0.202** | 0.162 | 0.223 | 0.157 | 0.215 | 0.173 | 0.241 | 0.154 | 0.243 |
| | 192 | 7.6% | **0.205** | **0.249** | 0.216 | 0.259 | 0.214 | 0.258 | 0.225 | 0.263 | 0.233 | 0.265 |
| | 336 | 6.8% | **0.262** | **0.289** | 0.279 | 0.291 | 0.275 | 0.292 | 0.289 | 0.305 | 0.282 | 0.293 |
| | 720 | 3.0% | **0.344** | **0.342** | 0.355 | 0.341 | 0.349 | 0.351 | 0.366 | 0.369 | 0.348 | 0.362 |
| 1st Count | | | 33 | | 3 | | 3 | | 1 | | 0 | |

performance (with 14 best scores), TFPS consistently achieves lower error in datasets with higher distribution shifts, such as ETTh2 and ETTm2, indicating its advantage in handling pattern heterogeneity.

These results reinforce the effectiveness of TFPS's pattern-specific modeling strategy, especially in scenarios where traditional large-scale foundation models struggle to generalize.

Table 10: Comparison between TFPS and MoE-based methods. The best results are highlighted in **bold** and the second best are underlined.

| Model | | IMP. | TFPS (Our) | | MoLE (2024) | | MoU (2024) | | KAN4TSF (2024) | |
|---|---|---|---|---|---|---|---|---|---|---|
| Metric | | MSE | MSE | MAE | MSE | MAE | MSE | MAE | MSE | MAE |
| ETTh1 | 96 | -4.3% | 0.398 | 0.413 | 0.383 | **0.392** | **0.381** | 0.403 | 0.382 | 0.400 |
| | 192 | 1.7% | **0.423** | **0.423** | 0.434 | 0.426 | 0.429 | 0.430 | 0.430 | 0.426 |
| | 336 | 1.6% | **0.484** | **0.461** | 0.489 | 0.478 | 0.488 | 0.463 | 0.498 | 0.467 |
| | 720 | 8.2% | **0.488** | **0.476** | 0.602 | 0.545 | 0.499 | 0.484 | 0.494 | 0.479 |
| ETTh2 | 96 | 10.4% | **0.313** | **0.355** | 0.413 | 0.360 | 0.317 | 0.358 | 0.318 | 0.358 |
| | 192 | 10.3% | **0.405** | **0.410** | 0.525 | 0.416 | 0.409 | 0.414 | 0.419 | 0.414 |
| | 336 | 7.1% | **0.392** | **0.415** | 0.423 | 0.434 | 0.397 | 0.420 | 0.447 | 0.452 |
| | 720 | 8.4% | **0.410** | **0.433** | 0.453 | 0.458 | 0.412 | 0.434 | 0.477 | 0.476 |
| ETTm1 | 96 | 13.5% | **0.327** | **0.367** | 0.338 | 0.380 | 0.465 | 0.442 | 0.333 | 0.371 |
| | 192 | 10.6% | **0.374** | **0.395** | 0.388 | 0.403 | 0.483 | 0.455 | 0.384 | 0.399 |
| | 336 | 11.8% | **0.401** | **0.408** | 0.417 | 0.431 | 0.540 | 0.488 | 0.407 | 0.413 |
| | 720 | 7.3% | **0.479** | **0.456** | 0.486 | 0.472 | 0.583 | 0.509 | 0.483 | 0.469 |
| ETTm2 | 96 | 13.9% | **0.170** | **0.255** | 0.238 | 0.271 | 0.179 | 0.263 | 0.175 | 0.260 |
| | 192 | 3.8% | **0.235** | **0.296** | 0.247 | 0.305 | 0.243 | 0.303 | 0.244 | 0.305 |
| | 336 | 3.3% | **0.297** | **0.335** | 0.308 | 0.343 | 0.306 | 0.343 | 0.308 | 0.347 |
| | 720 | 13.7% | **0.401** | **0.397** | 0.583 | 0.419 | 0.405 | 0.404 | 0.405 | 0.404 |
| 1st Count | | | 30 | | 1 | | 1 | | 0 | |

# G Compared with Other Methods

## G.1 Compared with Normalization Methods

In this section, we provide the detailed experimental results of the comparison between TFPS and five state-of-the-art normalization methods for non-stationary time series forecasting: SIN [17], SAN [34], Dish-TS [10], and RevIN [20]. We summarize the forecasting results of TFPS and baseline models in Table 8 and Table 9. Specifically, the results of FEDformer combined with SIN are taken from [17], while those of FEDformer with other normalization-based methods are reported by [34]. For a fair comparison, we additionally rerun all experiments for DLinear combined with each normalization method.

Table 8 and Table 9 presents the forecasting performance across all prediction lengths for each dataset, along with the relative improvements of TFPS over existing methods. As shown, TFPS consistently achieves the best performance in the majority of settings, demonstrating its strong adaptability to distributional and conceptual drifts in time series data.

We attribute this improvement to the accurate identification of pattern groups and the provision of specialized experts for each group, thereby avoiding the over-stationarization problem often associated with normalization methods.

## G.2 Compared with MoE-based Methods

As shown in Table 10, unlike MoE-based methods that rely on the Softmax function as a gating mechanism, our approach constructs a pattern recognizer to assign different experts to handle distinct patterns. This results in TFPS achieving relative improvements of 2.3%, 9.0%, 10.6%, and 9.1% across the four datasets, respectively.

## G.3 Compared with Distribution Shift Methods

As shown in Table 11, we compare with the methods for distribution shift. This results in TFPS achieving relative improvements of 6.7%, 6.6%, 4.8%, and 5.9% across the four datasets, respectively.

# H Model Analysis

**Detailed Results on the Number of Experts.** We provide the full results on the number of experts for the ETTh1 and Weather dataset in Figure 7.

Table 11: Comparison between TFPS and methods for Distribution Shift. The best results are highlighted in **bold** and the second best are underlined.

| Model | | IMP. | TFPS (Our) | | Koopa (2024)) | | SOLID (2024) | | OneNet (2024) | |
|---|---|---|---|---|---|---|---|---|---|---|
| Metric | | MSE | MSE | MAE | MSE | MAE | MSE | MAE | MSE | MAE |
| ETTh1 | 96 | 7.9% | 0.398 | 0.413 | **0.385** | 0.407 | 0.440 | 0.439 | 0.425 | **0.402** |
| | 192 | 10.3% | **0.423** | **0.423** | 0.445 | 0.434 | 0.492 | 0.466 | 0.452 | 0.443 |
| | 336 | 4.9% | **0.484** | **0.461** | 0.489 | 0.460 | 0.525 | 0.481 | 0.492 | 0.482 |
| | 720 | 4.4% | **0.488** | **0.476** | 0.497 | 0.480 | 0.517 | 0.496 | 0.504 | 0.496 |
| ETTh2 | 96 | 10.6% | **0.313** | **0.355** | 0.318 | 0.360 | 0.318 | 0.359 | 0.382 | 0.362 |
| | 192 | 4.7% | 0.405 | 0.410 | **0.378** | **0.398** | 0.414 | 0.418 | 0.435 | 0.426 |
| | 336 | 4.8% | **0.392** | **0.415** | 0.415 | 0.430 | 0.398 | 0.421 | 0.426 | 0.419 |
| | 720 | 6.8% | **0.410** | **0.433** | 0.445 | 0.456 | 0.424 | 0.441 | 0.456 | 0.437 |
| ETTm1 | 96 | 6.8% | **0.327** | **0.367** | 0.329 | 0.359 | 0.329 | 0.370 | 0.374 | 0.392 |
| | 192 | 2.0% | **0.374** | 0.395 | 0.380 | 0.393 | 0.379 | 0.400 | 0.385 | 0.435 |
| | 336 | 8.7% | **0.401** | **0.408** | 0.401 | 0.411 | 0.405 | 0.412 | 0.473 | 0.458 |
| | 720 | 2.0% | 0.479 | 0.456 | **0.475** | **0.448** | 0.482 | 0.464 | 0.496 | 0.483 |
| ETTm2 | 96 | 5.3% | **0.170** | **0.255** | 0.179 | 0.261 | 0.175 | 0.258 | 0.184 | 0.274 |
| | 192 | 3.8% | **0.235** | **0.296** | 0.246 | 0.305 | 0.241 | 0.302 | 0.248 | 0.384 |
| | 336 | 3.4% | **0.297** | **0.335** | 0.310 | 0.348 | 0.303 | 0.342 | 0.313 | 0.374 |
| | 720 | 9.0% | **0.401** | **0.397** | 0.405 | 0.402 | 0.456 | 0.436 | 0.425 | 0.438 |
| 1st Count | | | 25 | | 6 | | 0 | | 1 | |

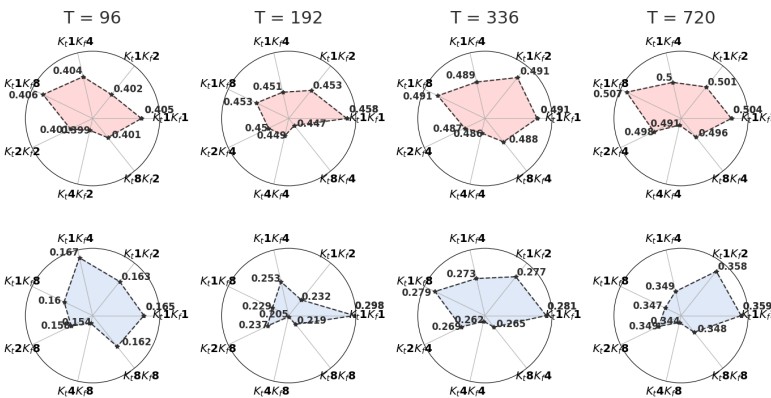

Figure 7: Results of expert number experiments for ETTh1 and Weather.

In Figure 7, we set the learning rate to 0.0001 and conducted four sets of experiments on the ETTh1 and Weather datasets, $K_t = 1$, $K_f = \{1, 2, 4, 8\}$, to explore the effect of the number of frequency experts on the results. For example, $K_t 1 K_f 4$ means that the TFPS contains 1 time experts and 4 frequency experts. We observed that $K_t 1 K_f 2$ outperformed $K_t 1 K_f 4$ in most cases, suggesting that increasing the number of experts does not always lead to better performance.

In addition, we conducted three experiments based on the optimal number of frequency experts to verify the impact of varying the number of time experts on the results. As shown in Figure 7, the best results for ETTh1 were obtained with $K_t 4 K_f 2$, $K_t 8 K_f 4$, $K_t 4 K_f 4$, $K_t 4 K_f 4$, while for Weather, the optimal results were achieved with $K_t 4 K_f 8$, $K_t 4 K_f 8$, $K_t 4 K_f 4$ and $K_t 4 K_f 8$. Combined with the average Wasserstein in Table 5, we attribute this to the fact that, in cases where concept drift is more severe, such as Weather, more experts are needed, whereas fewer experts are sufficient when the drift is less severe.

**Comparing Inter- and Intra-Cluster Differences via Wasserstein Distance.** To assess the effectiveness of the PI module, we replace it with a linear layer and compare the resulting inter- and intra-cluster Wasserstein distance heatmaps in Figure 8. The diagonal elements represent the average Wasserstein distances of patches within the same clusters. If these values are small, it indicates that the difference of patches within the same cluster is relatively similar. The off-diagonal elements represent

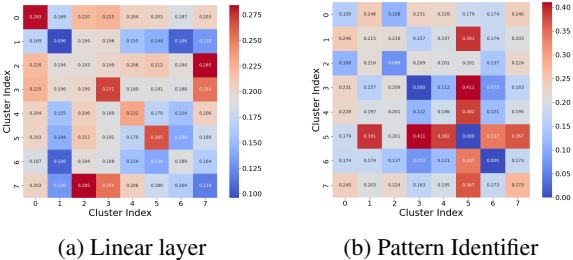

| (a) Linear layer | (b) Pattern Identifier |

Figure 8: Heatmap showing the Wasserstein distances of inter- and intra-cluster patches on ETTh1.

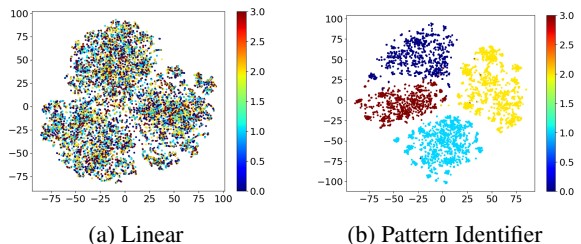

| (a) Linear | (b) Pattern Identifier |

Figure 9: Visualization of the embedded representations with t-SNE on ETTh1 for the time domain with $H = 96$. (a) t-SNE visualization with a Linear Layer replacing the Patch Identifier for comparison. (b) t-SNE visualization of TFPS.

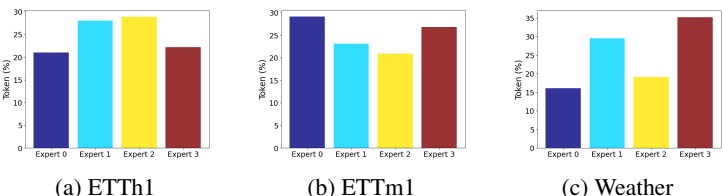

| (a) ETTh1 | (b) ETTm1 | (c) Weather |

Figure 10: MoE Expert allocation distributions of TFPS: the x-axis corresponds to the 4 experts, and the y-axis shows the proportion of tokens assigned to each expert.

the average Wasserstein distances between patches from different clusters, where larger values mean significant differences between the clusters. We observe that when using PI, the intra-cluster drift is smaller, while the inter-cluster shift is more pronounced compared to the linear layer. This indicates that our identifier effectively classifies and distinguishes between different patterns.

**TFPS produces differentiated token embeddings by adapting to the characteristics of the data.** Figure 9 presents the t-SNE visualization of the learned embedded representation on the ETTh1 for the time domain with $H = 96$. In the Figure 9 (a), where the pattern identifier is replaced with a linear layer, the representation lacks clear clustering structures, resulting in scattered and indistinct groupings. In contrast, Figure 9 (b) shows the visualization of the representation learned by the proposed method, which effectively captures discriminative features and reveals significantly clearer clustering patterns.

**TFPS implements dataset-specific token embeddings assignment in a data-driven way, effectively improving performance.** Figure 10 visualizes the expert allocation distributions across various datasets. Notably, ETTh1 and ETTm1 exhibit a high degree of consistency in their expert assignments, underscoring the model's ability to capture shared patterns. Conversely, the Weather dataset shows a distinctly different allocation pattern, highlighting the method's sensitivity to unique dataset characteristics and its capability to tailor expert assignments accordingly.

Table 12: The GPU memory (MB) and speed (inference time) of each model.

|  | TFPS | TSLANet | FITS | iTransformer | TFDNet-IK | PatchTST | TimesNet | DLinear | FEDformer |
|---|---|---|---|---|---|---|---|---|---|
| MSE | **0.423** | 0.448 | 0.445 | 0.441 | 0.458 | 0.460 | 0.441 | 0.434 | 0.441 |
| GPU Memory (MB) | 9.643 | 0.481 | 0.019 | 3.304 | 0.246 | 0.205 | 2.345 | 0.142 | 62.191 |
| Average Inference Time (ms) | 6.114 | 0.063 | 1.184 | 2.571 | 98.266 | 4.861 | 12.306 | 0.659 | 136.130 |

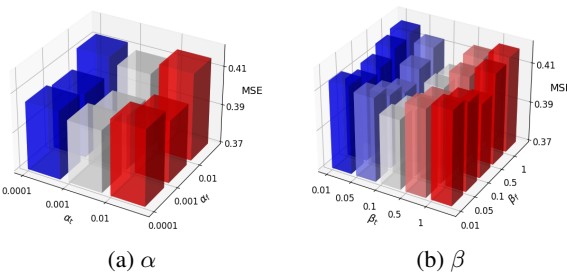

(a) $\alpha$          (b) $\beta$

Figure 11: Parameter sensitivity of $\alpha$ and $\beta$ of the proposed method on the ETTh1-96 dataset.

# I  Efficiency Analysis

To make this clearer, we present the results of ETTh1 for a prediction length of 192 from Table 1 and include additional results on runtime and computational complexity in Table 12. Due to the sparsity of MoPE, TFPS achieves a balance between performance and efficiency:

**Performance Superiority**. TFPS achieves an MSE of 0.423, outperforming TSLANet (0.448), FITS (0.445), PatchTST (0.460), and FEDformer (0.441). This represents a 5.6% improvement over TSLANet and a 8.0% improvement over PatchTST, highlighting its significant accuracy gains. While DLinear achieves an MSE of 0.434, TFPS still demonstrates a 2.5% relative improvement, making it the most accurate model among all baselines.

**Efficiency Gains**. TFPS maintains competitive runtime and memory efficiency.

- Runtime: TFPS runs in 6.457 ms, making it 2.8× faster than PatchTST (17.851 ms) and 11.2× faster than TimesNet (72.196 ms).
- Memory Usage: TFPS uses 9.643 MB of GPU memory, significantly less than FEDformer (62.191 MB) and comparable to iTransformer (3.304 MB). This makes TFPS suitable for resource-constrained applications while maintaining superior performance.

**Balancing Trade-offs**. While lightweight models like DLinear (0.434 MSE, 0.789 ms runtime) are slightly more efficient, TFPS delivers a performance improvement of 2.5%, providing a well-rounded solution that balances accuracy and efficiency effectively.

# J  Hyperparameter Sensitivity

In this section, we analysis the impact of the hyperparameters $\alpha$ and $\beta$ on the performance.

Specifically, we performed a grid search to optimize the hyperparameters $\alpha_t = \{0.0001, 0.001, 0.01\}$ and $\alpha_f = \{0.0001, 0.001, 0.01\}$, as shown in Figure 11 (a). After extensive testing, we ultimately fixed at $\alpha_t = \alpha_f = 10^{-3}$ in our experiments.

In addition, we conducted a grid search to optimize the balance factors $\beta_t = \{0.01, 0.05, 0.1, 0.5, 1\}$ and $\beta_f = \{0.01, 0.05, 0.1, 0.5, 1\}$. The performance under different parameter values is displayed in Figure 11 (b), from which we have the following observations:

- Firstly, the performance is affected when the value of $\beta$ is too low, indicating that the proposed clustering objective plays a crucial role in distinguishing patterns.
- Second, an excessive $\beta$ also has a negative on the performance. One plausible explanation is that the excessive value influences the learning of the inherent structure of original data, resulting in a perturbation of the embedding space.

Table 13: In the table, w/ Imaginary indicates that we incorporate both the real and imaginary parts into the network.

| | ETTh1 | | | | ETTh2 | | | |
|---|---|---|---|---|---|---|---|---|
| | 96 | 192 | 336 | 720 | 96 | 192 | 336 | 720 |
| TFPS | 0.398 | **0.423** | **0.484** | 0.488 | 0.313 | **0.405** | 0.392 | 0.410 |
| w/ Imaginary | **0.397** | 0.424 | 0.487 | **0.486** | **0.312** | 0.406 | **0.391** | **0.399** |

Table 14: Ablation study of PI components. The model variants in our ablation study include the following configurations across both time and frequency branches: (a) inclusion of the Time PI; (b) inclusion of the Frequency PI; (c) exclusion of both. The best results are in **bold**.

| Time PI | Frequency PI | ETTh1 | | | | ETTh2 | | | |
|---|---|---|---|---|---|---|---|---|
| | | 96 | 192 | 336 | 720 | 96 | 192 | 336 | 720 |
| ✓ | ✓ | **0.398** | **0.423** | **0.484** | **0.488** | **0.313** | **0.405** | **0.392** | **0.410** |
| ✓ | ✗ | 0.404 | 0.454 | 0.490 | 0.503 | 0.322 | 0.413 | 0.410 | 0.425 |
| ✗ | ✓ | 0.405 | 0.456 | 0.493 | 0.509 | 0.324 | 0.415 | 0.412 | 0.430 |
| ✗ | ✗ | 0.407 | 0.458 | 0.497 | 0.513 | 0.328 | 0.418 | 0.419 | 0.435 |

- Overall, we recommend setting $\beta$ around 0.1 for optimal performance.

# K   Full Ablation

## K.1   Impacts of Real/Imaginary Parts

To further validate the robustness of our approach, we adopted similar operations in FreTS to conduct experiments incorporating both the real and imaginary parts. The results in the Table 13 show that the performance of TFPS with the real part only is very similar to that when both parts are included, while requiring fewer parameters. This further reinforces the conclusion that TFPS remains highly effective even when focusing solely on the real part of the Fourier transform.

## K.2   Ablation on PI

The PI module plays a crucial role in identifying and characterizing distinct patterns within the time series data, while the gating network dynamically selects the most relevant experts for each segment. This collaborative mechanism allows the model to specialize in handling different patterns and adapt effectively to distribution shifts, thus mitigating the overfitting risks that arise from treating all data equally.

To validate the importance of PI empirically, we have conducted the ablation experiments comparing the model's performance by replacing the PI module with a linear layer in the Table 2 of main text. In addition, we supplement some ablation experiments in Table 14 to further verify the effectiveness of PI.

## K.3   Ablation on $R_1$ and $R_2$

We conducted ablation experiments to further verify the important roles of $R_1$ and $R_2$, as shown in Table 15.

## K.4   Replace MoPE with Alternative Designs

Here we provide the complete results of alternative designs for TFPS.

As show in Table 16, we have conducted addition experiments where we replaced the MoPE module with weighted multi-output predictor and stacked self-attention layers, keeping all other components

Table 15: Ablation study of Loss Constraint. The model variants in our ablation study include the following configurations across both time and frequency branches: (a) inclusion of the $R_1$; (b) inclusion of the $R_2$; (c) exclusion of both. The best results are in **bold**.

| $R_1$ | $R_2$ | ETTh1 | | | | ETTh2 | | | |
|---|---|---|---|---|---|---|---|---|---|
| | | 96 | 192 | 336 | 720 | 96 | 192 | 336 | 720 |
| ✓ | ✓ | **0.398** | **0.423** | **0.484** | **0.488** | **0.313** | **0.405** | **0.392** | **0.410** |
| ✓ | ✗ | 0.408 | 0.449 | 0.500 | 0.498 | 0.320 | 0.418 | 0.415 | 0.429 |
| ✗ | ✓ | 0.403 | 0.434 | 0.493 | 0.491 | 0.316 | 0.413 | 0.405 | 0.418 |
| ✗ | ✗ | 0.412 | 0.456 | 0.509 | 0.503 | 0.328 | 0.425 | 0.420 | 0.435 |

Table 16: Multi-output predictor and a stacked attention layer are used to replace MoPE in ETTh1 and ETTh2 datasets.

| | ETTh1 | | | | ETTh2 | | | |
|---|---|---|---|---|---|---|---|---|
| | 96 | 192 | 336 | 720 | 96 | 192 | 336 | 720 |
| TFPS | **0.398** | **0.423** | **0.484** | **0.488** | **0.313** | **0.405** | **0.392** | **0.410** |
| Multi-output Predictor | 0.403 | 0.435 | 0.492 | 0.491 | 0.317 | 0.407 | 0.399 | 0.425 |
| Attention Layers | 0.399 | 0.452 | 0.492 | 0.508 | 0.334 | 0.407 | 0.409 | 0.451 |

and configurations identical. The results demonstrate that our proposed method significantly outperforms them, which validates the importance of the Top-K selection and pattern-aware design in enhancing the model's representation capacity. In contrast, multi-output predictor and self-attention typically treats all data points uniformly, which may limit its ability to capture subtle distribution shifts or evolving patterns across patches.

## L  Algorithm of TFPS

We provide the pseudo-code of TFPS in Algorithm 1.

## M  Broader Impact

**Real-world Applications.** TFPS addresses the crucial challenge of time series forecasting, which is a valuable and urgent demand in extensive applications. Our method achieves consistent state-of-the-art performance in four real-world applications: electricity, weather, exchange rate, illness. Researchers in these fields stand to benefit significantly from the enhanced forecasting capabilities of TFPS. We believe that improved time series forecasting holds the potential to empower decision-making and proactively manage risks in a wide array of societal domains.

**Academic Research.** TFPS draws inspiration from classical time series analysis and stochastic process theory, contributing to the field by introducing a novel framework with the assistance pattern recognition. This innovative architecture and its associated methodologies represent significant advancements in the field of time series forecasting, enhancing the model's ability to address distribution shifts and complex patterns effectively.

**Model Robustness.** Extensive experimentation with TFPS reveals robust performance without exceptional failure cases. Notably, TFPS exhibits impressive results and maintains robustness in datasets with distribution shifts. The pattern identifier structure within TFPS groups the time series into distinct patterns and adopts a mixture of pattern experts for further prediction, thereby alleviating prediction difficulties. However, it is essential to note that, like any model, TFPS may face challenges when dealing with unpredictable patterns, where predictability is inherently limited. Understanding these nuances is crucial for appropriately applying and interpreting TFPS's outcomes.

Our work only focuses on the scientific problem, so there is no potential ethical risk.

**Algorithm 1** Time-Frequency Pattern-Specific architecture - Overall Architecture.

---

**Input**: Input lookback time series $X \in \mathbb{R}^{L \times C}$; input length $L$; predicted length $H$; variables number $C$; patch length $P$; feature dimension $D$; encoder layers number $n$; random Gaussian distribution-initialized subspace $\mathbf{D} = [\mathbf{D}^{(1)}, \mathbf{D}^{(2)}, \cdots, \mathbf{D}^{(K)}]$, each $\mathbf{D}^{(j)} \in \mathbf{R}^{q \times d}$, where $q = C \times D$ and $d = q/K$. Technically, we set $D$ as 512, $n$ as 2.

**Output**: The prediction result $\hat{Y}$.

1: $X = X.\texttt{transpose}$             $\triangleright X \in \mathbb{R}^{C \times L}$
2: $X_{PE} = \texttt{Patch}\,(X) + \texttt{Position Embedding}$     $\triangleright X_t^0 \in \mathbb{R}^{C \times N \times D}$
3: $\triangleright$ Time Encoder.
4: $X_t^0 = X_{PE}$
5: **for** $l$ **in** $\{1, \ldots, n\}$**:**
6:  $X_t^{l-1} = \texttt{LayerNorm}\,(X_t^{l-1} + \texttt{Self-Attn}\,(X_t^{l-1}))$.    $\triangleright X_t^{l-1} \in \mathbb{R}^{C \times N \times D}$
7:  $X_t^l = \texttt{LayerNorm}\,(X_t^{l-1} + \texttt{Feed-Forward}\,(X_t^{l-1}))$.    $\triangleright X_t^l \in \mathbb{R}^{C \times N \times D}$
8: **End for**
9: $z_t = X_t^l$               $\triangleright z_t^l \in \mathbb{R}^{C \times N \times D}$
10: $\triangleright$ Pattern Identifier for Time Domain.
11: $s_t = \texttt{Subspace affinity}\,(z_t, \mathbf{D})$    $\triangleright$ Eq. 6 of the paper $s_t \in \mathbb{R}^{C \times N \times D}$
12: $\hat{s}_t = \texttt{Subspace refinement}\,(s_t)$    $\triangleright$ Eq. 7 of the paper $\hat{s}_t \in \mathbb{R}^{C \times N \times D}$
13: $\triangleright$ Mixture of Temporal Pattern Experts.
14: $G(s) = \texttt{Softmax}\,(\texttt{TopK}\,(s_t))$
15: $h_t = \sum_{k=1}^{K} G(s)\texttt{MLP}_k(z_t)$    $\triangleright$ Eq. 10 and Eq. 11 of the paper $h_t \in \mathbb{R}^{C \times N \times D}$
16: $\triangleright$ Frequency Encoder.
17: $X_f^0 = X_{PE}$       $\triangleright$ Eq. 2 of the paper $X_f^0 \in \mathbb{R}^{C \times N \times P}$
18: **for** $l$ **in** $\{1, \ldots, n\}$**:**
19:  $X_f^{l-1} = \texttt{LayerNorm}\,(X_f^{l-1} + \texttt{Fourier}\,(X_f^{l-1}))$.    $\triangleright X_f^{l-1} \in \mathbb{R}^{C \times N \times D}$
20:  $X_f^l = \texttt{LayerNorm}\,(X_f^{l-1} + \texttt{Feed-Forward}\,(X_f^{l-1}))$.    $\triangleright X_f^l \in \mathbb{R}^{C \times N \times D}$
21: **End for**
22: $z_f = X_f^l$               $\triangleright z_f^n \in \mathbb{R}^{C \times N \times D}$
23: $\triangleright$ Pattern Identifier for Frequency Domain.
24: $s_f = \texttt{Subspace affinity}\,(z_f, \mathbf{D})$    $\triangleright$ Eq. 6 of the paper $s_f \in \mathbb{R}^{C \times N \times D}$
25: $\hat{s}_f = \texttt{Subspace refinement}\,(s_f)$    $\triangleright$ Eq. 7 of the paper $\hat{s}_f \in \mathbb{R}^{C \times N \times D}$
26: $\triangleright$ Mixture of Frequency Pattern Experts.
27: $G(s) = \texttt{Softmax}\,(\texttt{TopK}\,(s_f))$
28: $h_f = \sum_{k=1}^{K} G(s)\texttt{MLP}_k(z_f)$    $\triangleright$ Eq. 10 and Eq. 11 of the paper $h_f \in \mathbb{R}^{C \times N \times D}$
29: $h = \texttt{Concat}(h_t, h_f)$          $\triangleright h \in \mathbb{R}^{C \times N \times 2*D}$
30: **for** $c$ **in** $\{1, \ldots, C\}$**:**
31:  $\hat{Y} = \texttt{Linear}\,(\texttt{Flatten}\,(h))$.    $\triangleright$ Project tokens back to predicted series $\hat{Y} \in \mathbb{R}^{C \times H}$
32: **End for**
33: $\hat{Y} = \hat{Y}.\texttt{transpose}$            $\triangleright \hat{Y} \in \mathbb{R}^{H \times C}$
34: **Return** $\hat{Y}$       $\triangleright$ Output the final prediction $\hat{Y} \in \mathbb{R}^{H \times C}$

# N  Limitations

Though TFPS demonstrates promising performance on the benchmark dataset, there are still some limitations of this method. First, the patch length is primarily chosen heuristically, and the current design struggles with handling indivisible lengths or multi-period characteristics in time series. While this approach works well in experiments, it lacks generalizability for real-world applications. Second, the real-world time series data undergo expansion, implying that the new patterns continually emerge over time, such as an epidemic or outbreak that had not occurred before. Therefore, future work will focus on developing a more flexible and automatic patch length selection mechanism, as well as an extensible solution to address these evolving distribution shifts.

