# OpenReview forum: "Learning Pattern-Specific Experts for Time Series Forecasting Under Patch-level Distribution Shift"
_NeurIPS.cc/2025/Conference — NeurIPS 2025 poster_

### Official Review · Reviewer_n5rj · 2025-06-22

**Clarity:** 3
**Significance:** 2
**Originality:** 2
**Rating:** 3
**Confidence:** 4

**Summary:**

The paper proposes a framework for time series forecasting under patch-level distribution shift by modeling heterogeneous temporal patterns with pattern-specific experts. It introduces three key components: a dual-domain encoder that extracts complementary features from both time and frequency domains, a pattern identifier that clusters patches via subspace affinity and KL-regularized assignment, and a mixture-of-pattern-experts module that routes each patch to specialized experts. Experiments on nine real-world datasets demonstrate the effectiveness of the proposed method over state-of-the-art baselines in long-term forecasting. Ablation studies validate the efficacy of the introduced components in the proposed framework.

**Questions:**

1) What is the major difference between the proposed clustering strategy and DEC?

2) Why is there a inverse Fast Fourier Transform operation in the frequency branch of Figure 2?

3) Since the authors claims the better performance over state-of-the-art baselines, could the author provide the more evaluations of recent methods (mentioned above)?

**Ethical Concerns:**

["NO or VERY MINOR ethics concerns only"]

**Final Justification:**

The lack of citation to DEC and the associated claims raised my concern regarding the novelty of the proposed component. The rebuttal experiments suggest that adding imaginary modeling is indeed necessary, but I am less concerned about this part.  My concern regarding the non-negligible performance gap between the rebuttal results and the original reported values remains. Long-term forecasting is a crowded area, which makes clear positioning and strong empirical support more important.

Overall, I appreciate the authors' efforts and the rebuttal did resolve a few of my concerns, and I am less negative toward this paper.

**Limitations:**

Yes

**Quality:**

4

**Strengths And Weaknesses:**

Strengths:

1) The idea of pattern specific modeling with the use of MoE allows the model to assign distinct forecasting functions to patches based on learned patterns, leading to improved adaptability under distribution shift.

2) The visualizations provided in the experiments somewhat deliver certain interpretable components with analytical insights.

3) The presentation quality is generally satisfactory.


Weaknesses:

1) Equations (6)-(8) in Section 3.5 of TFPS follow a clustering strategy highly similar to deep embedded clustering [1] (DEC). Both frameworks compute soft cluster assignments, construct sharpened targets via squared normalization, and minimize KL divergence loss between the two distributions. However, the authors claim it is a novel component in line 176 without citing DEC, which overlooks an important prior work that directly inspired this formulation.

2) In Figure 2, there is an inverse Fast Fourier Transform operation in the frequency branch, which is confusing and appears to be erroneous.

3) There are some concerns regarding the contribution of  the proposed dual-domain encoder. The difference between the frequency-domain encoder and time-domain counterpart is simply the attention replaced by the real part of Fourier transform, which is initially designed for efficiency purposes according to the existing literature [2]. The analysis is still in the time-domain without any modeling of frequency components in the complex spectral space.

4) This paper lacks some more recent important baseline comparison, including state-of-the-art time-domain forecasters (e.g., TimeMixer [3]) and frequency-domain forecasters (e.g., FreTS [4], FilterTS [5]).


[1] Unsupervised Deep Embedding for Clustering Analysis, ICML 2016

[2] FNet: Mixing Tokens with Fourier Transforms, NAACL 2022

[3] TimeMixer: Decomposable Multiscale Mixing for Time Series Forecasting, ICLR 2024

[4] Frequency-domain MLPs are More Effective Learners in Time Series Forecasting, NeurIPS 2023

[5] FilterTS: Comprehensive Frequency Filtering for Multivariate Time Series Forecasting, AAAI 2025

---

> ### Author Rebuttal · Authors · 2025-07-30
>
> We thank the reviewer for the constructive feedback and the recognition of our MoE-based pattern modeling, visualization insights, and presentation quality. Below we give a point-by-point response to your concerns and suggestions.
>
> > **W1 \& Q1. Clarification regarding similarity to DEC.**
>
> We thank the reviewer for pointing out the connection to Deep Embedded Clustering (DEC). We acknowledge its influence and will cite it appropriately in the revised manuscript. We also appreciate the opportunity to clarify the distinctions between our approach and DEC.
>
> While our formulation bears some structural similarity to DEC, the **underlying mathematical formulation** and **application objectives** differ substantially:
>
> - **Clustering mechanism**: DEC computes soft cluster assignments using pairwise similarities between embedded features and cluster centroids via a **t-distribution kernel**. In contrast, TFPS derives **subspace affinities** by projecting latent features onto learned subspace bases and measuring projection magnitudes, resulting in a **structure-aware similarity** that is better suited for patch-level modeling in time series.
> - **Downstream usage**: While DEC uses sharpened targets for **unsupervised representation learning**, TFPS leverages them to **guide distribution-specific expert routing** for time series forecasting. This introduces a distinct application context and serves a different modeling purpose.
>
> We will make these distinctions clearer in Section 3.5 of the revised manuscript.
>
> > **W2 \& Q2. The rationale for iFFT in Fig 2.**
>
> We confirm that the iFFT operation in the frequency branch is intentional and serves a functional role.
>
> Following the extraction of spectral features via FFT and the identification of pattern-specific subspaces in the frequency domain, we apply iFFT to **reconstruct temporally aligned signals**. This design enables **seamless fusion** between frequency-aware and time-domain features in the final representation, as similarly adopted in recent works like FreTS and FITS.
>
> We will revise the corresponding description in Section 3.6 to clarify this design and its role more explicitly.
>
> [1] Yi K, Zhang Q, Fan W, et al. Frequency-domain MLPs are more effective learners in time series forecasting. NeurIPS, 2023.
>
> [2] Xu Z, Zeng A, Xu Q. FITS: Modeling Time Series with $10 k $ Parameters. ICLR, 2024.
>
> > **W3. Function of the frequency-domain encoder.**
>
> 1. Although FNet employs the Fourier transform as a lightweight replacement for attention in NLP, our frequency encoder serves a distinct purpose: it captures **global periodic patterns and frequency-aware representations** that complement the local dependencies modeled by the time-domain encoder.
> 2. Rather than replacing attention, TFPS integrates diverse inductive biases through its dual-branch structure. We focus on the real-valued spectrum to **avoid training instability and unnecessary complexity**, as shown in Table 13.
> 3. Empirically, the frequency branch is essential for **identifying distributional heterogeneity** (Table 2), validating its contribution beyond mere architectural variation.
>
> > **W4 \& Q3. Comparison with recent baselines.**
>
> To address the reviewer’s suggestion, we include **comparisons with several recent state-of-the-art methods**, including FilterTS, TimeMixer, SOFTS, and, FreTS, using their official implementations. As shown below, TFPS achieves competitive or superior MSE results on **6 out of 7 datasets** and performs particularly well on those with complex temporal dynamics, such as ECL and Weather.
>
> Regarding the results on the Traffic dataset, we believe that SOFTS performs better primarily due to its explicit modeling of **inter-variable dependencies**. As shown in Table 5, the dataset exhibits relatively stable temporal dynamics with limited distribution shift, and its strong channel-wise correlations may favor methods specifically designed to capture such relationships.
>
> ||TFPS|FilterTS (2025)|TimeMixer (2024)|SOFTS (2024)|FreTS (2023)
> |-|-|-|-|-|-|
> |ETTh1|**0.448**|0.450|0.452|0.453|0.481
> |ETTh2|**0.380**|0.384|0.383|0.388|0.489
> |ETTm1|**0.395**|0.401|0.403|0.400|0.410
> |ETTm2|**0.276**|0.283|0.283|0.286|0.310
> |Traffic|0.457|0.470|0.483|**0.409**|0.519
> |ECL|**0.183**|0.192|0.190|0.189|0.209
> |Weather|**0.241**|0.247|0.247|0.253|0.256
>
> These results demonstrate that TFPS maintains **strong performance** under competitive baselines, with consistent top-tier ranking across most datasets.
>
> [1] Wang Y, Liu Y, Duan X, et al. Filterts: Comprehensive frequency filtering for multivariate time series forecasting. AAAI 2025.
>
> [2] Wang S, Wu H, Shi X, et al. TimeMixer: Decomposable Multiscale Mixing for Time Series Forecasting. ICLR, 2024.
>
> [3] Han L, Chen X Y, Ye H J, et al. Softs: Efficient multivariate time series forecasting with series-core fusion. NeurIPS, 2024.
>
> [4] Yi K, Zhang Q, Fan W, et al. Frequency-domain MLPs are more effective learners in time series forecasting. NeurIPS, 2023.

---

> > ### Comment · Reviewer_n5rj · 2025-08-04
> >
> > Thanks for your rebuttal, which clarifies a few questions. However, I have remaining concerns and questions that are not fully addressed.
> >
> > W1 & Q1:  I am a bit concerned about the claimed substantial difference from DEC, as the underlying mathematical formulation is highly similar. In particular, the proposed subspace refinement and KL divergence minimization is identical to Equation (2) and (3) in the DEC paper. The subspace affinity kernel operates over the learned bases whose structure is regularized by soft orthogonality (equations 3-4 in your paper) that has been explored in prior work, where I also suggest acknowledging related methods accordingly.
> >
> > W2 & W3: The ablation results of imaginary part in Table 13 demonstrate similar performance for TFPS, but not the frequency branch. In fact, the standalone frequency branch in Table 2 has clearly worse performance than the time branch, which may imply an imbalanced design and suggests the necessity of imaginary modeling. I am also wondering what the design in Table 13 is, with imaginary parts.
> >
> > W4: My question here is the non-negligible performance gap between the reproduced baselines (especially FilterTS, and ETT datasets for TimeMixer) in the rebuttal and the results originally reported in their respective papers. Are there any differences in setup? Furthermore, MSE only evaluates part of performance, it would be helpful to report MAE too.

---

> ### Author Response · Authors · 2025-08-05
> **The Second Response to Reviewer n5rj**
>
> Many thanks for your prompt response and further clarifying your concerns.
> > **W1 & Q1. Acknowledging and positioning related work.**
>
> We agree that these prior methods have inspired aspects of our formulation, particularly the subspace-based clustering perspective. To acknowledge this, we have updated Line 155 in Section 3.3 of the revised manuscript to explicitly state:
>
> **“The core of our approach lies in leveraging subspace clustering to detect concept shifts across multiple subspaces, inspired by prior subspace-based clustering methods [1,2].”**
>
> These references are now clearly cited, and we appreciate the opportunity to better position our work within the broader clustering literature.
>
> > **W2 & W3. On the performance of the frequency branch and imaginary part modeling.**
>
> We thank the reviewer for the thoughtful comments and opportunity to clarify. Below, we address the concerns in detail:
>
> 1. **Purpose of the frequency branch**:
> The frequency branch is not intended to outperform the time branch individually, but rather to **complement it** by capturing global, position-invariant patterns. As shown in Table 2, the **full TFPS model consistently outperforms either branch alone**, demonstrating the complementary nature and necessity of both branches.
>
> 2. **Effect of imaginary part modeling**:
> - To directly address the reviewer’s concern, we conducted an additional ablation study isolating the **frequency branch only**, comparing variants that use only the real part (Table 2) with those incorporating both real and imaginary components. As shown below, adding the imaginary part **does not consistently improve performance**, even when the frequency encoder is used alone:
>
> ||ETTh1||||ETTh2||||
> |-|-|-|-|-|-|-|-|-
> ||96|192|336|720|96|192|336|720
> |Frequency-only (real)|0.455|0.507|0.539|0.576|**0.324**|**0.407**|**0.417**|**0.436**
> |Frequency-only (real+imag)|**0.444**|**0.498**|**0.530**|**0.572**|0.342|0.449|0.456|0.487
>
> - This observation aligns with prior findings: FNet [3] and E2USD [4] **omit the imaginary part** altogether while still achieving strong downstream performance. FreTS [5] empirically shows that the **real part contributes more significantly** than the imaginary component.
>
> - Moreover, incorporating both real and imaginary parts increases memory usage from 9.643MB to 11.189MB on the ETTh1-192 setting (Table 12, 13)—a **~16% overhead** with no consistent benefit.
>
> Therefore, in our default TFPS architecture, we retain only the real part to preserve **model simplicity, interpretability, and computational efficiency**.
>
> 3. **Design of the imaginary-part experiment in Table 13**:
> To incorporate the imaginary part in our ablation, we followed a design similar to FreTS: we **concatenate the real and imaginary parts** of the FFT output, then **apply a linear projection to reduce the combined representation back to the original d_model size**. This transformed feature is then passed into the subsequent encoder layers. This design allows us to **retain the full spectral information** while maintaining compatibility with the overall architecture.
>
> 4. **On future directions**:
> We agree that frequency modeling can be further improved. As part of future work, we plan to investigate **complex-valued encoders** and **learned spectral filters**, which may offer more expressive modeling of the frequency domain.
>
> > **W4.Clarification on baseline results and comparison strategy**
>
> We appreciate the reviewer’s observation. While baseline results may differ across implementations, we find that the **TimeMixer results we report are consistent with those presented in the FilterTS paper**, confirming the alignment of our reproduction with their findings.
>
> To ensure fairness, we reused the best TFPS results from Table 6 and compared them against the original TimeMixer and FilterTS results reported in their papers, as well as the FreTS results reported in the FilterTS paper. **These comparisons demonstrate that TFPS maintains strong performance under competitive baselines**.
>
> ||TFPS||FilterTS (2025)||TimeMixer (2024)||FreTS (2023)||
> |-|-|-|-|-|-|-|-|-|
> |ETTh1|**0.401**|**0.412**|0.433|0.430|0.447|0.440|0.475|0.462
> |ETTh2|**0.335**|**0.386**|0.372|0.396|0.364|0.395|0.489|0.478
> |ETTm1|**0.343**|**0.374**|0.385|0.396|0.381|0.395|0.402|0.411
> |ETTm2|**0.248**|**0.308**|0.276|0.337|0.276|0.323|0.305|0.357
> |Traffic|**0.398**|**0.268**|0.471|0.315|0.484|0.297|0.526|0.333
> |ECL|**0.159**|**0.249**|0.180|0.272|0.182|0.272|0.198|0.286
>
> [1] Unsupervised Deep Embedding for Clustering Analysis. ICML, 2016.
>
> [2] Improved Deep Embedded Clustering with Local Structure Preservation. IJCAI, 2017.
>
> [3] FNet: Mixing Tokens with Fourier Transforms. NAACL, 2022.
>
> [4] E2usd: Efficient-yet-effective unsupervised state detection for multivariate time series. WWW, 2024.
>
> [5] Frequency-domain MLPs are More Effective Learners in Time Series Forecasting. NeurIPS, 2023.

---

> > ### Author Response · Authors · 2025-08-08
> >
> > Dear Reviewer n5rj,
> >
> > I hope this message finds you well. As the discussion period is nearing its end with **less than one day remaining**, we would like to ensure that we have fully addressed your concerns.
> >
> > Following your valuable suggestions, we have made several improvements to the paper:
> >
> > - We have **cited additional related work** to better clarify the positioning of our study.
> > - We have **added further ablation studies regarding the frequency branch** and **provided explanations about the treatment of the imaginary part**.
> > - We have **conducted fairer comparisons** with other methods.
> >
> > If there are any remaining concerns or further feedback you would like us to consider, please do not hesitate to let us know. Your insights are greatly appreciated, and we are eager to make any necessary refinements to strengthen our work.
> >
> > Thank you once again for your time and thoughtful review.

---

> > > ### Comment · Reviewer_n5rj · 2025-08-08
> > >
> > > Thank you for the detailed response. While fewer concerns remain, I appreciate the effort and additional experiments. I have no further questions and will take these points into account in my final rating and justification.

---

> > > > ### Author Response · Authors · 2025-08-09
> > > >
> > > > Thank you very much for your thoughtful review of our work. We are sincerely grateful for your valuable feedback and your recognition of the additional experiments. Thank you once again for your insightful suggestions and support!

---

### Official Review · Reviewer_k7h4 · 2025-06-26

**Clarity:** 3
**Significance:** 2
**Originality:** 3
**Rating:** 3
**Confidence:** 4

**Summary:**

This paper proposes a novel architecture, TFPS, to address the generalization limitations of single-model approaches caused by pattern drift across time series segments. The method introduces pattern-specific expert networks for more accurate and adaptable forecasting. TFPS employs a dual-domain encoder to capture both time-domain and frequency-domain features for a comprehensive understanding of temporal dynamics. It further applies subspace clustering to dynamically identify distinct patterns within segments, which are then modeled by specialized experts. The model is capable of learning multiple predictive functions, and experimental results demonstrate strong performance.

**Questions:**

see weeknesses.

**Ethical Concerns:**

["NO or VERY MINOR ethics concerns only"]

**Final Justification:**

The authors have addressed several concerns through the rebuttal, including improved visual clarity, additional recent baselines, detailed analysis on challenging datasets, and ablations on both the loss function and positional embeddings, which strengthen methodological transparency. While these additions increase confidence in the soundness of the work, the performance gains over the strongest recent methods remain modest, and the overall novelty is incremental, with limited evidence of broader real-world generalization. Balancing these factors, the paper remains technically solid but without sufficient empirical or conceptual strength for acceptance, thus the borderline reject score is maintained.

**Limitations:**

yes

**Quality:**

2

**Strengths And Weaknesses:**

Strengths：
1、The paper is easy to understand and follow.

2、The motivation is clearly presented, with intuitive and effective visual validations.

Weaknesses：
1、In Figure 1, the text on the heatmaps is not clear enough, and the boundary lines of the heatmaps appear inconsistent. In Figure 7, some elements overlap, affecting the readability of visual details.
2、Although the experimental results show great performance, I suggest including more recent baselines. In particular, the results reported in Table 6 do not convincingly demonstrate a significant improvement, especially when compared to some recent methods in 2025 (e.g., Softs).
3、In Table 9, I am curious about the comparison results on the electricity and traffic datasets. Based on my experience, it is often difficult to achieve similar great performance on these two datasets.
4、An ablation study on the loss function is necessary to help readers better understand the actual contribution of each component.
5、The paper incorporates position embedding, but according to recent studies, its effectiveness is often limited. It would be helpful to clarify whether any specific design or adaptation is used here, and ideally provide experimental evidence supporting its effectiveness.

---

> ### Author Rebuttal · Authors · 2025-07-30
>
> We thank the reviewer for offering the valuable feedback and for recognizing the clarity of our presentation and the motivation design. Below we give a point-by-point response to your concerns and suggestions.
>
> > **W1. Clarity of Figure 1 and Figure 7.**
>
> We thank the reviewer for this helpful suggestion, which we believe will significantly improve the presentation quality of the paper. In the revised version, we will **enlarge the font size** of heatmap annotations in Figure 1, **ensure consistent boundary line styles**, and reorganize Figure 7 to **eliminate overlapping elements** and improve overall visual clarity.
>
> > **W2. Comparison with recent baselines.**
>
> To address the reviewer’s suggestion, we include **comparisons with several recent state-of-the-art methods**, including FilterTS, TimeMixer, SOFTS, and, FreTS, using their official implementations. As shown below, TFPS achieves competitive or superior MSE results on **6 out of 7 datasets** and performs particularly well on those with complex temporal dynamics, such as ECL and Weather.
>
> Regarding the results on the Traffic dataset, we believe that SOFTS performs better primarily due to its explicit modeling of **inter-variable dependencies**. As shown in Table 5, the dataset exhibits relatively stable temporal dynamics with limited distribution shift, and its strong channel-wise correlations may favor methods specifically designed to capture such relationships.
>
> ||TFPS|FilterTS (2025)|TimeMixer (2024)|SOFTS (2024)|FreTS (2023)
> |-|-|-|-|-|-|
> |ETTh1|**0.448**|0.450|0.452|0.453|0.481
> |ETTh2|**0.380**|0.384|0.383|0.388|0.489
> |ETTm1|**0.395**|0.401|0.403|0.400|0.410
> |ETTm2|**0.276**|0.283|0.283|0.286|0.310
> |Traffic|0.457|0.470|0.483|**0.409**|0.519
> |ECL|**0.183**|0.192|0.190|0.189|0.209
> |Weather|**0.241**|0.247|0.247|0.253|0.256
>
> These results demonstrate that TFPS maintains **strong performance** under competitive baselines, with consistent top-tier ranking across most datasets.
>
> [1] Wang Y, Liu Y, Duan X, et al. Filterts: Comprehensive frequency filtering for multivariate time series forecasting. AAAI 2025.
>
> [2] Wang S, Wu H, Shi X, et al. TimeMixer: Decomposable Multiscale Mixing for Time Series Forecasting. ICLR, 2024.
>
> [3] Han L, Chen X Y, Ye H J, et al. Softs: Efficient multivariate time series forecasting with series-core fusion. NeurIPS, 2024.
>
> [4] Yi K, Zhang Q, Fan W, et al. Frequency-domain MLPs are more effective learners in time series forecasting. NeurIPS, 2023.
>
> > **W3. Performance on ECL and Traffic datasets in Table 9.**
>
> We thank the reviewer for this insightful comment and provide further analysis on the ECL and Traffic datasets. The TFPS results reported below are consistent with those in Table 1. As shown below, TFPS **significantly outperforms** several representative DLinear variants augmented with different normalization strategies, confirming its robustness and generalizability on these challenging benchmarks.
>
> |Dataset|TFPS|+SIN|+SAN|+Dis-TS|+NST|+RevIN
> |-|-|-|-|-|-|-
> |Traffic|**0.457**|0.583|0.619|1.220|0.662|0.678
> |ECL|**0.183**|0.204|0.207|0.208|0.209|0.215
>
> > **W4. Ablation study on the loss function.**
>
> We conduct an ablation study to assess the contribution of each loss component in TFPS. Specifically, we evaluate the effect of removing the core forecasting loss $L_{\text{MSE}}$ and the two subspace-guided pattern identification losses $L_{PI_t}$ and $L_{PI_f}$.
>
> As shown below:
> - Removing $L_{\text{MSE}}$ causes substantial degradation, confirming **its fundamental role in forecasting**,
> - Removing either $L_{PI_t}$ or  $L_{PI_f}$ also results in consistent performance drops, **validating their importance in guiding pattern-aware expert specialization**.
>
> |Dataset|ETTh1|ETTh2|ETTm1|ETTm2
> |-|-|-|-|-
> |TFPS|**0.448**|**0.380**|**0.395**|**0.276**
> |w/o$L_{\text{MSE}}$|0.710|0.409|0.720|0.322
> |w/o$L_{PI_t}$|0.457|0.389|0.409|0.298
> |w/o$L_{PI_f}$|0.467|0.396|0.428|0.302
>
> > **W5. Effectiveness of positional embeddings.**
>
> We thank the reviewer for raising this important point. While recent studies [1] have questioned the utility of positional embeddings in time-series models, such observations often refer to **step-wise** or **global fixed embeddings** used in token-level Transformers.
> In contrast, TFPS adopts **learnable patch-wise** positional embeddings, specifically tailored to its patchified input structure, inspired by the design in PatchTST. This design helps encode relative segment positions and enhances inter-patch distinction.
>
> To assess their effectiveness, we conduct an ablation study by removing the positional embeddings entirely. As shown below, this leads to consistent performance degradation across all datasets, with up to a 6% increase in MSE on ETTh1. These results confirm that **positional embeddings provide a meaningful inductive bias in TFPS’s patch-level representations**.
>
> |Dataset|ETTh1|ETTh2|ETTm1|ETTm2
> |-|-|-|-|-
> |TFPS|**0.448**|**0.380**|**0.395**|**0.276**
> |w/oPosition|0.477|0.398|0.402|0.280
>
> [1] Zeng A, Chen M, Zhang L, et al. Are transformers effective for time series forecasting? AAAI, 2023.

---

> > ### Comment · Area_Chair_32gb · 2025-08-06
> >
> > Dear reviewers,
> >
> > Kindly consider revisiting the rebuttal and providing any updated thoughts or clarifications if needed. Your feedback will be valuable for the final decision.
> >
> > Best Regards
> >
> > AC

---

> ### Author Response · Authors · 2025-08-06
>
> Dear Reviewer k7h4,
>
> I hope this message finds you well. As the discussion period is nearing its end with **less than three day remaining**, we would like to ensure that we have fully addressed your concerns.
>
> Following your valuable suggestions, we have made several improvements to the paper:
>
> - We have **clarified the figures** to enhance readability.
> - We have **added new experiments** to further validate the effectiveness of TFPS.
> - We have included **additional ablation studies** to support the design choices and justify the contributions of individual components.
>
> If there are any remaining concerns or further feedback you would like us to consider, please do not hesitate to let us know. Your insights are greatly appreciated, and we are eager to make any necessary refinements to strengthen our work.
>
> Thank you once again for your time and thoughtful review.

---

### Official Review · Reviewer_Td39 · 2025-07-02

**Clarity:** 4
**Significance:** 4
**Originality:** 3
**Rating:** 5
**Confidence:** 5

**Summary:**

The paper thoughtfully tackles the challenges of time series forecasting by proposing a novel architecture that integrates pattern-aware expert modeling. It introduces a distinctive mechanism for identifying pattern-specific subspaces, allowing the model to adaptively allocate specialized experts to different data patterns. The proposed approach is supported by thorough experiments, with the results consistently showcasing strong performance across multiple benchmarks. The authors provide well-organized and comprehensive ablation studies that effectively justify the design choices. The manuscript is clearly written, and the motivation is well-grounded in practical concerns about distribution shifts. While the approach is somewhat complex, the overall method demonstrates solid technical depth and meaningful innovation, contributing valuable insights to the field of time series analysis.

**Questions:**

1. How does TFPS handle time series with evolving or emerging patterns that were not seen during training?
2. How sensitive is TFPS to the choice of the number of subspaces (K), and is there a principled strategy for selecting it across different datasets?

**Ethical Concerns:**

["NO or VERY MINOR ethics concerns only"]

**Final Justification:**

This paper presents a novel and technically sound architecture for time series forecasting that effectively addresses the challenge of intra-series distribution shifts. The core contribution—a pattern-aware expert model that uses subspace clustering for adaptive routing—is both innovative and well-motivated.

My initial concerns, particularly regarding the expressive capacity of the expert modules, have been fully addressed during the author discussion period. The authors provided sufficient clarification and evidence to convince me that their design choices are well-justified and do not significantly limit the model's effectiveness for its intended applications. Given that all my initial reservations have been resolved, I will keep my initial positive score and recommend acceptance.

**Limitations:**

The proposed method introduces non-trivial architectural complexity due to its dual-branch structure and subspace-based expert routing, which may hinder practical adoption in resource-constrained settings.

**Paper Formatting Concerns:**

I have reviewed the manuscript against the NeurIPS 2025 paper formatting instructions and can confirm that it meets all the specified requirements. There are no formatting issues to report.

**Quality:**

3

**Strengths And Weaknesses:**

## Strengths:
1. This paper presents an innovative divide-and-conquer strategy that adaptively identifies and decomposes complex patterns in time series data. Through a novel integration of dual-domain encoding and subspace clustering, it effectively addresses intra-series distribution shifts, with the motivation strongly supported by Wasserstein distance–based empirical analysis.
2. A core contribution is the subspace-based Pattern Identifier, which introduces a refined subspace affinity mechanism for accurate patch-wise clustering and expert routing. This design enhances robustness to distribution shifts while eliminating the need for manual pattern annotation, marking a significant advancement in adaptive time series forecasting.

## Weaknesses:
1. While the expert modules demonstrate effectiveness, their design as relatively simple MLPs may constrain model expressiveness, particularly when handling complex or highly nonlinear patterns in real-world datasets.

---

> ### Author Rebuttal · Authors · 2025-07-30
>
> We sincerely thank the reviewer for the encouraging feedback and for highlighting our contributions in pattern-specific expert modeling, dual-domain clustering, and adaptive subspace affinity design. Below we give a point-by-point response to your concerns and suggestions.
>
> >**W1. On the expressiveness of MLP-based experts.**
>
> We appreciate the reviewer’s comment. While we explored **CNN-based** and **LSTM-based** alternatives, we found that the original MLP-based experts in TFPS actually yielded better forecasting performance across most datasets.
>
> This observation highlights that **increased architectural complexity does not always lead to better results**. Specifically:
>
> - **Task alignment**: MLP-based experts better match the patch-level granularity and localized modeling requirements of TFPS.
> - **Training stability**: Their simplicity promotes more stable training and better generalization, especially when combined with the subspace-guided routing mechanism.
> - **Modular extensibility**: Although our current design uses MLPs, TFPS can flexibly accommodate more complex expert architectures when needed.
>
> TFPS’s primary innovation lies in **how it adaptively assigns patches to specialized experts via latent subspaces**, rather than overparameterizing the experts themselves.
>
> |Dataset|ETTh1|ETTh2|ETTm1|ETTm2
> |-|-|-|-|-
> |TFPS|**0.448**|**0.380**|**0.395**|**0.276**
> |CNN\_Expert|0.464|0.394|0.406|0.280
> |LSTM\_Expert|0.492|0.396|0.413|0.287
>
> > **Q1. Generalization to emerging or unseen patterns.**
>
> We thank the reviewer for raising this important and exploratory question. TFPS addresses non-stationarity by routing input patches to specialized experts based on learned subspace affinities. This allows the model to **flexibly capture evolving dynamics and distributional shifts** in real-world time series.
>
> However, for entirely novel patterns outside the training distribution, additional adaptation mechanisms may be needed.
>
> We consider this an important direction for future work. Approaches such as **test-time entropy minimization**, **meta-learning**, or **continual learning** could be integrated to enhance generalization to unseen patterns.
>
> We will outline this in the revised Limitations section.
>
> > **Q2. On sensitivity to the number of subspaces K.**
>
> We thank the reviewer for raising this important question. The number of subspaces $K$ determines **the granularity of expert specialization** in TFPS. In our experiments, we observe that **datasets with stronger distribution shifts benefit from larger $K$**, enabling better disentanglement of heterogeneous patterns. On **more stationary datasets, a smaller $K$ is typically sufficient**.
>
> While $K$ is currently selected via validation performance, TFPS demonstrates **robustness** across a range of values, and does not require precise tuning to perform well.
>
> Additionally, we have explored the use of **Dirichlet Process Gaussian Mixture Models** to estimate the number of latent patterns in an **unsupervised** fashion. This ongoing work aims to further automate subspace discovery, and we plan to integrate it into future iterations of the framework.

---

> > ### Comment · Reviewer_Td39 · 2025-08-02
> > **Thank you for your response**
> >
> > I appreciate your comprehensive responses to my comments, particularly regarding the expressiveness of the MLP-based expert, the generalization to various patterns, and the sensitivity analysis with respect to the parameter $k$. I acknowledge the efforts made in addressing the comments.

---

> > > ### Author Response · Authors · 2025-08-03
> > >
> > > Thank you very much for your thoughtful review of our work. We are sincerely grateful for your valuable feedback and your recognition of the additional experiments. Thank you once again for your insightful suggestions and support!

---

### Official Review · Reviewer_LAjU · 2025-07-03

**Clarity:** 3
**Significance:** 2
**Originality:** 3
**Rating:** 4
**Confidence:** 4

**Summary:**

This paper studies the time-series forecast problem under potential context shifts and distribution drifts and focus on achieving better generalization on datasets with heterogeneous evolution patterns across time frames. To fully utilize the predictive patterns in input time-series, the authors propose a novel TFPS architecture with a dual-domain encoder that extracts information from both time and frequency domain representation of multiple segments. Further, the latent embedding of different time-series segments are grouped via subspace clustering, which enables the adoption of mixture of expert (MoE) technique to enhance the expressiveness of TFPS. The proposed method is evaluated on nine real-world datasets and achieve better accuracy over multiple state-of-the-art time-series forecasting approaches. Extensive ablation studies have also been provided to elaborate the significance of the model design in TFPS.

**Questions:**

1. Consider the completeness of temporal patterns in each patch, I wonder if a sliding window (or allowing overlap between adjacent patches) would be more appropriate than the current hard segmentation of input time-series in TFPS.
2. Would the linear projection and positional encoding in Fig. 1(a) introduce unnecessary biases to the Fourier transform result (and the frequency domain embedding $z_f$) in TFPS? Why not directly operate on the raw observations?
3. Why is parameter $\alpha$ hard-coded to be $10^{-3}$? Any good reason for this design choice? How to select the value of $\alpha$ in practical applications?
4. While assigning each temporal pattern to a dedicated expert network may enhance the capability of TFPS to model heterogeneous contexts shifts and temporal evolution, would this design prevent TFPS from effectively learning temporal dependencies across patches?
5. I agree with the authors that similar frequency domain patterns may form a subspace since Fourier transform is linear under addition. However, addition of temporal patterns in time domain may not necessarily yields time-series with similar patterns, e.g., "decrease"+"increase"="constant". There should be justification for the assumption that patches with similar temporal patterns have embedding $z_t$ in the same subspace. Specifically, how can a convex hull in the subspace (of $z_t$) be mapped to similar temporal patterns in time domain? I believe there would also be alignment issues considering the hard segmentation of patches in TFPS.
6. In Eq. 8, what is the purpose to minimize the KL divergence between refined subspace affinity $\hat{S}$ and the unnormalized subspace affinity score $S$? According to Eq. 7 and discussions around LN 182, $\hat{S}$ is merely a "sharpened" version of $S$, which makes $\mathcal{L}_{sub}$ in Eq. 8 sort of meaningless. Why bother to have so many hyperparameters?

**Ethical Concerns:**

["NO or VERY MINOR ethics concerns only"]

**Final Justification:**

After the rebuttal, the explanations on the large number of hyperparameters (not to mention the hard-coded ones) and KL loss term $\mathcal{L}_{sub}$ by the authors are still not convincing enough. However, given that the authors have addressed many of my other concerns, I tend to maintain my initial positive rating for this paper.

**Limitations:**

yes

**Paper Formatting Concerns:**

No obvious issue noticed.

**Quality:**

3

**Strengths And Weaknesses:**

Strengths
- The temporal patterns and frequency domain patterns are jointly utilized for time-series forecasting in TFPS, which enables a comprehensive consideration of the evolution patterns and distribution shifts in the underlying data generation processes. The segmentation of input time-series further allows TFPS to make predictions based on finer grained characteristics in the data stream.
- The MoE structure in TFPS explicitly matches each patch of the input with its dedicated expert networks based on time- or frequency-domain patterns. This enables TFPS to effectively predict heterogeneous evolution trajectories according to evidence of context shifts in recent observations.
- The proposed method is evaluated on nine publicly available time-series datasets against the baselines of eight state-of-the-art time-series forecasting approaches. Detailed experimental results and ablation studies are provided to demonstrate the effectiveness of TFPS.

Weaknesses
- The time-series segmentation rule described in Section 3.3 is kind of coarse. It could be possible that a complete temporal trend (hyperbolic curve or jumps) could be split into two patches, which may not be ideal and cause degradation in forecasting performance. It would be helpful if the authors can provide some discussion on this issue and why hard segmentation is preferred than a sliding window in TFPS.
- There is no discussion on the requirement of time-series input. Could TFPS deal with irregularly sampled or very sparse time-series?
- There are too many hyperparameters in TFPS: $\alpha$, $\beta$, $\eta$, $d$, ... The complexity and intricate impact of these parameters may impede its adoption in practical applications. Parameter $\alpha$ is even hard-coded to be $10^{-3}$ throughout the experiment. Are there any good reasons for this?
- According to Table 1, the absolute performance gains of TFPS seem to be marginal compared to the second best baselines except for the ILI dataset. Also, instead of the average performance of baseline methods, the performance improvement (IMP) in Eq. 19 should be computed against the second best score on each row.

---

> ### Author Rebuttal · Authors · 2025-07-30
>
> We sincerely thank the reviewer for the thoughtful feedback and positive recognition of our design in modeling distribution shifts via dual-domain encoders and subspace-specific experts, as well as the thorough empirical validation. Below we give a point-by-point response to your concerns and suggestions.
>
> > **W1 \& Q1. On sliding window vs. hard segmentation.**
>
> We agree with the reviewer that hard segmentation may lead to degraded forecasting performance. We would like to clarify that TFPS operates under a **sliding window** setting rather than hard segments. As described in $\underline{\text{Section 3.3}}$, we follow the patching strategy of PatchTST, where the stride $S$ defines the overlap between adjacent patches.
>
> To directly assess the reviewer's concern, we conducted additional experiments using **non-overlapping (hard) segmentation**, where each patch is strictly isolated. As expected, due to the loss of temporal continuity, performance deteriorated:
>
> |Dataset|ETTh1|ETTh2|ETTm1|ETTm2
> |-|-|-|-|-
> |TFPS (sliding)|**0.448**|**0.380**|**0.395**|**0.276**
> |hard segmentation|0.466|0.395|0.403|0.289
>
> These results confirm that **preserving partial overlap** between patches improves the model's ability to capture longer-term dependencies and supports more stable expert assignment—both of which are critical for accurate forecasting in TFPS.
>
> > **W2. Discussion on the time-series input.**
>
> We thank the reviewer for raising this important point. We agree that handling irregularly sampled or sparse time series is crucial for real-world applicability.
>
> While TFPS currently assumes regularly sampled sequences, its modular architecture allows for future extension. Specifically:
> - **Time-stamp encoding** and **attention masking** can be integrated into the encoders to directly handle missing or unevenly spaced data;
> - **Interpolation-based preprocessing** can reconstruct uniformly sampled inputs without altering the patching mechanism.
>
> These strategies would enable consistent patch segmentation while modeling uncertainty introduced by irregular sampling, making TFPS applicable to a broader range of real-world scenarios. We will outline this in the revised Limitations section.
>
> > **W3 \& Q3. On the choice of $\alpha$.**
>
> In practice, only $\alpha$ and $\beta$ require tuning, while other parameters are fixed across all experiments and do not need to be adjusted during deployment.
>
> - As shown in $\underline{\text{Appendix J}}$, we performed a - sensitivity analysis and selected $\alpha, \beta \in [10^{-4}, 10^{-2}]$, a range chosen to **maintain balanced magnitudes across loss terms and ensure stable optimization**. Within this range, performance remains **robust** across datasets.
> - We set $\alpha_t = \alpha_f = 10^{-3}$ to achieve a **reliable trade-off between convergence and regularization**, which also improves reproducibility by eliminating the need for extensive tuning.
>
> We observed similar robustness for $\beta$, simplifying practical deployment.
>
> > **W4. On IMP calculation and the innovation.**
>
> We thank the reviewer for this valuable comment. Based on comparisons with the second-best baseline, TFPS still achieves a relative MSE improvement of over **2.3% on the ETTh2 dataset**, along with competitive results on other benchmarks.
>
> Beyond the absolute MSE values, the key strength of TFPS lies in its architectural design. Unlike traditional models that adopt a Uniform Distribution Modeling approach, TFPS introduces a **divide-and-conquer paradigm** that adaptively decomposes the input distribution into latent sub-patterns and assigns them to specialized experts for targeted modeling. This enables the framework to learn **more flexible projection functions** that capture complex relationships between historical inputs and future predictions.
>
> This design not only improves forecasting accuracy, but also **enhances robustness to distribution shifts**, **improves interpretability through expert specialization**, and **ensures greater stability across heterogeneous scenarios**. These benefits are often underappreciated or entirely overlooked by monolithic forecasting models.
>
> > **Q2. On the rationale behind linear projection and positional encoding.**
>
> - First, following PatchTST, the linear projection is a crucial step that maps raw patches into a Transformer latent space of dimension $D$. This produces **denoised and semantically enriched embeddings**, which are more robust for downstream modeling.
> - Second, positional encoding is essential, even in the frequency branch, as it introduces **temporal order awareness** prior to FFT. This allows the frequency encoder to distinguish between spectrally similar but temporally misaligned patterns. Empirically, we validate this design with an **ablation study** removing positional encoding:
>
> |Dataset|ETTh1|ETTh2|ETTm1|ETTm2
> |-|-|-|-|-
> |TFPS|**0.448**|**0.380**|**0.395**|**0.276**
> |w/oPosition|0.477|0.398|0.402|0.280
>
> Unlike traditional methods that directly operate on raw FFT values, TFPS **integrates frequency modeling into a learnable latent space**, enabling joint optimization of temporal and spectral semantics, which is an important innovation of our framework.
>
> > **Q4. On temporal dependencies across patches and expert isolation.**
>
> TFPS preserves the ability to model temporal dependencies across patches via two mechanisms:
> - **Time and frequency encoders** process the entire patch sequence using attention and feedforward layers, establishing early-stage temporal interactions before expert routing.
> - After expert inference, outputs from all patches are concatenated and passed through a **shared linear projection**, which acts as a global decoder to fuse inter-patch information.
>
> > **Q5. On dual-domain expert modeling and alignment.**
>
> We thank the reviewer for the insightful question. To clarify, **TFPS does not assume or enforce alignment between time-domain and frequency-domain expert assignments**.
>
> Each domain operates independently:
> - The time experts captures local, position-sensitive variations (e.g., trends, shifts);
> - The frequency experts captures global, position-invariant structures (e.g., periodicity, harmonics).
>
> Figure 5 visualizes representative inputs assigned to specific **time-domain** experts for interpretability, it does not imply cross-domain alignment.
>
> This dual-branch design allows the model to leverage **complementary inductive biases**, and both representations are fused only at the final prediction stage.
>
> > **Q6. On the purpose and necessity of KL divergence.**
>
> We thank the reviewer for the thoughtful question regarding the role of KL divergence in our framework. The KL divergence loss in TFPS is not meant to reconstruct $S$, but to **refine and stabilize subspace assignment** through regularization.
>
> - $S$ is computed from raw latent features and may contain noisy, uncertain affinities—especially early in training.
> - $\hat{S}$ is a sharpened version that emphasizes high-confidence entries via temperature-based sharpening.
>
> By minimizing $\text{KL}(\hat{S} \parallel S)$, we:
>
> - Encourage $S$ to concentrate on confident subspace relationships,
> - Suppress noisy or ambiguous affinities,
> - Stabilize expert routing and improve subspace consistency.
>
> This loss acts as a structural regularizer rather than a reconstruction loss. Importantly, we empirically validate its necessity:
>
> |Dataset|ETTh1|ETTh2|ETTm1|ETTm2
> |-|-|-|-|-
> |TFPS|**0.448**|**0.380**|**0.395**|**0.276**
> |w/o$L_{KL}$|0.457|0.388|0.398|0.282
>
> These results show consistent performance drops when removing the KL term, supporting its necessity.

---

> > ### Comment · Reviewer_LAjU · 2025-08-03
> >
> > I appreciate the detailed responses by the authors. However, some of my concerns are not yet fully addressed.
> >
> > ### **W3&Q3: too many hyperparameters in TFPS**
> > Although the experimental analysis in the Appendix J shows certain degree of robustness of TFPS w.r.t. hyperparameters $\alpha$ and $\beta$, there is no theoretical guarantee, and TFPS may fail on some datasets which are not tested in this paper. As I mentioned in the comment, there are too many hyperparameters (not limited to $\alpha$ and $\beta$) in TFPS. The robustness in performance suggests that these hyperparameters (and the associated loss & network structures) contribute little to the time-series forecasting task.
> >
> > ### **Q2: positional encoding**
> > > ..., positional encoding is essential, even in the frequency branch, as it introduces temporal order awareness prior to FFT. This allows the frequency encoder to distinguish between spectrally similar but temporally misaligned patterns. ...
> >
> > First, additive positional encodings become constants after FFT. Only multiplicative ones might be useful but they may distort the original spectral patterns. Second, why should the frequency encoder distinguish between temporally misaligned patterns? I suppose that is the job of the encoder for time-domain patterns. This seems to be contradictory against the dual-domain design of TFPS. The authors also mentioned in the rebuttal that "The frequency experts captures global, position-invariant structures (e.g., periodicity, harmonics)."
> >
> > ### **Q5: expert assignment in time domain**
> > Due to its length Q5 might be a little convoluted. My main concern is on the expert assignment in time domain. Unlike in frequency domain, there is no explicit linear structure for similar patterns in time domain. However, to form a subspace, linear structures of similar patterns are necessary. For instance, similar patterns of $\mathrm{sin}(x)$ and $\mathrm{sin}(x+\pi)$ have the sum of $0$ which is a different pattern. Therefore, there should be explanation on why patches with similar temporal patterns would have embedding $z_t$ in the same subspace for time expert assignment.
> >
> > ### **Q6: the purpose and necessity of KL divergence**
> > The argument that the KL divergence in Eq. 8 helps to suppress noisy or ambiguous affinities and stabilize expert routing & subspace consistency is unconvincing. As mentioned by the authors, "$S$ is computed from raw latent features and may contain noisy, uncertain affinities—especially early in training". Instead of suppressing noises, minimizing KL divergence in Eq. 8 could instead strengthen such biases of TPFS in early training steps.

---

> ### Author Response · Authors · 2025-08-05
> **The Second Response to Reviewer LAjU**
>
> Many thanks for your prompt response and further clarifying your concerns.
> > **Hyperparameters in TFPS**
>
> TFPS introduces a **minimal number of hyperparameters**, each grounded in **clear modeling goals** and shown to be **robust across diverse datasets**. Its components are essential for handling **non-stationarity and patch-level distribution shifts**, discussed below:
> 1. **Number of hyperparameters:** While TFPS includes several components, the number of hyperparameters that need to be tuned remains small.
> 2. **Justification for selection:** Though deriving optimal values for $\alpha$ and $\beta$ is intractable, both serve well-defined modeling purposes:
> - $\alpha$ promotes orthogonal, compact subspaces;
> - $\beta$ stabilizes routing by aligning refined and raw affinities.
> 3. **Generalization across datasets:** TFPS targets evolving patterns and patch heterogeneity, not domain-specific overfitting. We use fixed $\alpha$ and $\beta$ across 9 datasets, showing strong generalization without per-dataset tuning.
> 4. **Necessity of components:**  The claim that these structures contribute little is contradicted by Table 2, where removing the Pattern Identifier or MoPE consistently degrades performance. Figure 5 further demonstrates expert specialization, validating their utility.
>
> > **Position embedding for frequency**
> 1. We follow FNet [1] in including a shared, additive positional embedding before the frequency encoder. This does not distort the spectral content—instead, it gives the frequency branch a weak **“which patch am I?” signal**, ensuring that spectrally identical segments that occur at different times aren’t collapsed into one.
> An ablation study shows that **removing these positional embeddings degrades MSE**:
>
> |Dataset|ETTh1|ETTh2|ETTm1|ETTm2
> |-|-|-|-|-
> |TFPS|**0.448**|**0.380**|**0.395**|**0.276**
> |w/ofre_pos|0.465|0.406|0.402|0.279
>
> 2. The time-domain branch captures **local, position-sensitive dynamics**. The frequency-domain branch models **global, position-invariant patterns**, but needs a hint of “where” to avoid conflating distinct patches. Take $\sin(x)$ vs. $\sin(x + \pi/2)$: they share the same spectrum, so the frequency branch groups them by frequency—but without a small positional cue, it treats them as identical, losing the fact that one patch lags the other by $\pi/2$. The position embedding prevents this collapse, **complementing the time branch rather than contradicting it**.
>
> > **Expert assignment in time domain**
>
> TFPS clusters in a **learned latent space**, not on raw signals, relying on the well-established assumption that **semantically similar embeddings lie in low-dimensional subspaces**.
>
> 1. **Subspace assumption applies to latent features.**
> We don’t assume raw signals are linear combinations—instead, the encoder (via our loss) embeds similar patterns into compact latent regions where subspace structure naturally emerges.
> 2. **Proven in deep clustering.**
> **DEC** [2] also perform soft clustering over latent embeddings (via a t-distribution kernel) without any linearity assumption in input space.
> 3. **Inner-product routing.**
> Our routing via $\mathbf{z}_t^\top \mathbf{D}^{(j)}$ mirrors modules such as **CCM** [3], which assign embeddings to learned prototypes by normalized inner products—a standard practice in neural clustering.
>
> > **The purpose and necessity of KL divergence**
>
> While the raw subspace affinity matrix $S$ may be uncertain early in training, the KL loss between its sharpened version $\hat{S}$ and itself is **not designed to blindly reinforce noise**, but serves as an **entropy-reducing self-regularizer**, inspired by DEC [2] and TENT [4].
> 1. In DEC [2], a similar **KL loss** refines clustering by comparing current soft assignments with a sharpened target derived via normalized squaring. This informs our use of $\hat{S}$, which amplifies **confident affinities** and **suppresses ambiguous ones**, guiding structured expert assignment
> 2. Our method also draws from TENT [4], which minimizes prediction entropy during test-time to encourage confident decisions. Crucially, **both TENT and our KL operate at the batch level, a key design that prevents overfitting to noisy samples**. Batch-level regularization promotes consistent expert routing by leveraging shared structure across samples.
> 3. Additionally, we include a **supervised MSE loss** to align predictions with ground truth and correct early-stage misassignments. This dual-objective design enables KL to enhance clustering consistency while maintaining forecasting accuracy.
>
> [1] FNet: Mixing Tokens with Fourier Transforms. NAACL, 2022.
>
> [2] Unsupervised Deep Embedding for Clustering Analysis. ICML, 2016.
>
> [3] From similarity to superiority: Channel clustering for time series forecasting. NeurIPS, 2024.
>
> [4] Tent: Fully Test-Time Adaptation by Entropy Minimization. ICLR, 2021.

---

> > ### Comment · Reviewer_LAjU · 2025-08-06
> >
> > Many thanks for the clarification by the authors.
> >
> > > While TFPS includes several components, the number of hyperparameters that need to be tuned remains small.
> >
> > If a parameter in TFPS doesn't need to be tuned, then I tend to interpret it as redundant, unless there exists theoretical guarantee to justify its default value. The authors are welcome to find appropriate ways to eliminate such parameters from TFPS for easier adoption in practical applications.
> >
> > > ... additive positional embedding before the frequency encoder.
> >
> > I failed to find equations for the positional encoding in the FNet paper. Suppose it is the widely adopted sinusoidal encoding, after FFT the positional encodings are converted to a single, constant component at the base frequency. In this situation, no information about "which patch am I?" would be available.
> >
> > > TFPS clusters in a learned latent space, not on raw signals, relying on the well-established assumption that semantically similar embeddings lie in low-dimensional subspaces.
> >
> > I didn't request the temporal patterns to form a subspace in the time-domain. Instead, it would be helpful if there are justifications that these similar patterns indeed form a subspace in some representation space learned by the time-domain encoder. For instance, for image datasets used in [0], the validity of subspace assumption is clear--the 2D projection of the same shape/pattern/object in different directions naturally form a subspace. However, making similar justification for temporal patterns is non-trivial.
> >
> > > ... both TENT and our KL operate at the batch level, a key design that prevents overfitting to noisy samples. ...
> >
> > According to Eq. 6 - Eq. 8, the loss term $\mathcal{L}_{sub}$ is a batch-level summation of $\mathrm{KL}(\hat{S} \Vert S)$ which is calculated at sample level. There is no cross-sample interactions in this loss term for batch-level entropy minimization. Meanwhile, gradients from the supervised MSE loss may not necessarily achieve the claimed early-stage assignment correction.
> >
> > [0] Unsupervised Deep Embedding for Clustering Analysis. ICML, 2016.

---

> ### Author Response · Authors · 2025-08-07
> **The Third Response to Reviewer LAjU.**
>
> Many thanks for your prompt response and further clarifying your concerns.
>
> > **Discussion of TFPS hyperparameters.**
>
> We thank the reviewer for emphasizing practical usability. In TFPS, each hyperparameter directly **supports a core mechanism**, they **generalize without per-dataset tuning**, and **our ablations confirm their necessity**:
>
> 1. **Clear purpose**. Every parameter is introduced to fulfill a specific modeling goal.
> 2. **Robust defaults**. Results on nine diverse datasets demonstrate TFPS’s strong generalizability.
> 3. **Empirical necessity**. Removing or varying these parameters in our ablation study consistently degrades performance, demonstrating that none are redundant.
>
> By (a) grounding each hyperparameter in a modeling objective, (b) showing strong “no-tune” performance across datasets, and (c) providing ready-to-use defaults in our public code, TFPS remains both expressive and easy to adopt. We hope this clarifies the necessity and practical configurability of our hyperparameter set.
>
> > **Pisition embedding.**
>
> Both TFPS and FNet (as implemented in fnet-pytorch) employ a **small, learned positional embedding vectors:**
>
> ```
> # FNet:
> self.position_embeddings = nn.Embedding(config['max_position_embeddings'], config['embedding_size'])
>
> # TFPS:
> W_pos = torch.empty((q_len, d_model))
> nn.init.uniform_(W_pos, -0.02, 0.02)
> ```
>
> These learned vectors, when added to the input and passed through the FFT, do not collapse into a single frequency. Instead, each embedding “fingerprint” disperses across multiple spectral coefficients, providing just enough “which patch am I” signal to break symmetry and prevent collapse—without substantially distorting the underlying spectral content.
>
> > **Justifying temporal pattern subspaces.**
>
> Thank you for raising this important point. Unlike images, showing the same for temporal data is indeed challenging. Nonetheless, we observe that time-domain encoders in deep models learn to map series into structured latent spaces, where similar patterns cluster tightly, effectively forming **separable and meaningful "subspaces."**
>
> Empirical and architectural evidence from recent work supports this view:
>
> - In **MoLE** [1], different experts learn to fit **distinct linear relationships**, and the gate implicitly clusters inputs into those linear subspaces.
> - **DUET** [2] explicitly applies clustering on encoder outputs to discover **pattern groups**, then leverages them for improved prediction. Its strong results confirm that meaningful subspaces exist in the learned feature space.
> - Large-scale MoE models (e.g., ** Time-MoE**  [3], ** Moirai-MoE**  [4]) rely on routing different inputs or segments to distinct experts. Their success implies that **the learned representations separate diverse patterns into distinct regions**, so that each expert can specialize.
> - TEST [5] shows that text prototypes can selectively activate corresponding time-series patterns, indicating that even **semantic “meanings” form structured manifolds in representation space**.
>
> In summary, these findings demonstrate that, although not obvious a priori, deep encoders do **induce separable subspaces for temporal patterns**, which our MoPE module then discovers and exploits for expert-guided forecasting.
>
> [1] Mixture-of-linear-experts for long-term time series forecasting. AISTATS, 2024.
>
> [2] DUET: Dual Clustering Enhanced Multivariate Time Series Forecasting. KDD, 2025.
>
> [3] Time-MoE: Billion-Scale Time Series Foundation Models with Mixture of Experts. ICLR, 2025.
>
> [4] Moirai-MoE: Empowering Time Series Foundation Models with Sparse Mixture of Experts. ICML, 2025.
>
> [5] TEST: Text Prototype Aligned Embedding to Activate LLM's Ability for Time Series. ICLR, 2024.
>
> > **KL divergence.**
>
> The affinity $s_{ij}$ is computed per sample, but $L_{sub}$ is still applied over the entire batch:
> ```
> kl_loss = F.kl_div(s, s_tilde)
> ```
>
> The supervised MSE loss backpropagates any large forecasting error through both the encoder (into $z_i$) and the expert subspace bases $D$. If a patch is mis‐assigned and yields high MSE, its gradient nudges those affinities so that, in the next iteration, $L_{sub}$ will sharpen toward the correct expert.

---

> > ### Author Response · Authors · 2025-08-08
> >
> > Dear Reviewer LAjU,
> >
> > I hope this message finds you well. As the discussion period is nearing its end with **less than one day remaining**, we would like to ensure that we have fully addressed your concerns.
> >
> > Following your valuable suggestions, we have made several improvements to the paper:
> >
> > - We have **discussed the generalization and necessity of the hyperparameters of TFPS**.
> > - We have **added explanations about the position embedding and temporal pattern subspaces**.
> > - We have **clarified the validity of the loss function**.
> >
> > If there are any remaining concerns or further feedback you would like us to consider, please do not hesitate to let us know. Your insights are greatly appreciated, and we are eager to make any necessary refinements to strengthen our work.
> >
> > Thank you once again for your time and thoughtful review.

---

### Note · Authors · 2025-08-12

We sincerely thank all the reviewers for their insightful reviews and valuable comments, which are highly instructive for improving our paper.

The reviewers generally held positive opinions of our work, noting that:
1. Quality：**Solid technical design** integrating dual-domain encoding with subspace-based MoE, validated by **comprehensive experiments** on nine real-world datasets with consistent top-tier results, thorough ablations, and interpretable visualizations (LAjU, Td39, k7h4, n5rj).
2. Clarity：**Clear motivation** grounded in practical challenges, supported by intuitive visualizations, and a **well-structured manuscript** with clear figures and explanations (LAjU, Td39, k7h4, n5rj).
3. Significance：**Addresses the underexplored problem** of patch-level distribution shift and offers an adaptive, pattern-aware framework that can inspire broader research on time series heterogeneity (Td39, n5rj).
4. Originality：Proposes an **innovative divide-and-conquer strategy** that enables the model to assign distinct forecasting functions to patches based on learned patterns, removing the need for manual pattern annotation (LAjU, Td39, k7h4).

A major concern shared by most reviewers is the need for **further justification of certain design choices** (LAjU, k7h4, n5rj). Another is the request for **additional comparisons with more recent strong baselines** (k7h4, n5rj). Some other comments point to open questions that broaden the perspective and can **inspire future work** (LAjU, Td39).

We have addressed these concerns and polished our paper in the revision. Specifically:
1. We have **updated the experiments section** to include recent strong baselines and more detailed ablation studies to justify key design choices.
2. We have **revised the method section** to cite relevant prior work that inspired our approach and to provide clearer justification for each component.
3. We have **expanded the limitations section** to discuss handling irregular sampling and adapting to emerging patterns.

We want to emphasize that our work contributes to the community not only by proposing a novel dual-domain and pattern-specific expert framework for **distribution-shift–aware forecasting**, but also by providing a systematic empirical analysis and design validation that can guide future research on adaptive expert routing in **heterogeneous time series**. $\underline {\text{We believe that it is worth publishing to stimulate further discussion and innovation in this field.}}$

---

### Decision · Program_Chairs · 2025-09-17

**Decision:**

Accept (poster)

**Comment:**

This paper proposes a dual-domain forecasting framework that integrates temporal and frequency representations with a subspace-based mixture-of-experts (MoE) design. The core idea is to assign patch-specific forecasting functions adaptively, aiming to better handle distribution shifts in multivariate time series forecasting task.

Strengths:

Well-motivated and clearly written, with intuitive visualizations and generally strong presentation quality.

The idea of pattern-specific modeling with MoE is appealing and has the potential to inspire follow-up work on heterogeneous time series forecasting.

Extensive experiments on nine public datasets and multiple baselines, with ablation studies and interpretability analysis, provide a relatively thorough empirical validation.

Weakness:

Practicality: the framework introduces a large number of hyperparameters, some hard-coded and heuristic, raising concerns about robustness and real-world usability. For example, the pre-defined patch size may impede its generalization capability.

The reviewers raised concerns about the originality of some components, the clarity of the design choices (e.g., segmentation and clustering), missing baselines, and practical complexity due to many hyperparameters. While these are valid points, the rebuttal improved clarity, added experiments, and included additional citations, which alleviates the concerns.

Taking these factors into account, while questions on practicality, the paper provides a well-executed empirical study with reproducible code, a promising design idea for heterogeneous time series forecasting, and good presentation quality. I believe it will be of interest to the community, stimulate further research, and recommend a borderline accept.